# Adaptive Negative Curvature Descent
# with Applications in Non-convex Optimization

**Mingrui Liu[†], Zhe Li[†], Xiaoyu Wang[‡], Jinfeng Yi[♮], Tianbao Yang[†]**
[†]Department of Computer Science, The University of Iowa, Iowa City, IA 52242, USA
[‡] Intellifusion [♮] JD AI Research
`mingrui-liu, tianbao-yang@uiowa.edu`

## Abstract

Negative curvature descent (NCD) method has been utilized to design deterministic or stochastic algorithms for non-convex optimization aiming at finding second-order stationary points or local minima. In existing studies, NCD needs to approximate the smallest eigen-value of the Hessian matrix with a sufficient precision (e.g., $\epsilon_2 \ll 1$) in order to achieve a sufficiently accurate second-order stationary solution (i.e., $\lambda_{\min}(\nabla^2 f(\mathbf{x})) \geq -\epsilon_2$). One issue with this approach is that the target precision $\epsilon_2$ is usually set to be very small in order to find a high quality solution, which increases the complexity for computing a negative curvature. To address this issue, we propose an adaptive NCD to allow an adaptive error dependent on the current gradient's magnitude in approximating the smallest eigen-value of the Hessian, and to encourage competition between a noisy NCD step and gradient descent step. We consider the applications of the proposed adaptive NCD for both deterministic and stochastic non-convex optimization, and demonstrate that it can help reduce the the overall complexity in computing the negative curvatures during the course of optimization without sacrificing the iteration complexity.

## 1 Introduction

In this paper, we consider the following optimization problem:

$$\min_{\mathbf{x} \in \mathbb{R}^d} f(\mathbf{x}), \tag{1}$$

where $f(\mathbf{x})$ is a non-convex smooth function with Lipschitz continuous Hessian, which could has some special structure (e.g., expectation structure or a finite-sum structure). A standard measure of an optimization algorithm is how fast the algorithm converges to an optimal solution. However, finding the global optimal solution to a generally non-convex problem is intractable [13] and is even a NP-hard problem [10]. Therefore, we aim to find an approximate second-order stationary point with:

$$\|\nabla f(\mathbf{x})\| \leq \epsilon_1, \quad \text{and} \quad \lambda_{\min}(\nabla^2 f(\mathbf{x})) \geq -\epsilon_2, \tag{2}$$

which nearly satisfy the second-order necessary optimality conditions, i.e., $\nabla f(\mathbf{x}_*) = 0, \lambda_{\min}(\nabla^2 f(\mathbf{x}_*)) \geq 0$, where $\|\cdot\|$ denotes the Euclidean norm and $\lambda_{\min}(\cdot)$ denotes the smallest eigen-value function. In this work, we refer to a solution that satisfies (2) as an $(\epsilon_1, \epsilon_2)$-second-order stationary solution. When the function is non-degenerate (i.e., strict saddle or the Hessian at all saddle points have a strictly negative eigen-value), then the solution satisfying (2) is close to a local minimum for sufficiently small $0 < \epsilon_1, \epsilon_2 \ll 1$. Please note that in this paper we do not follow the tradition of [14] that restricts $\epsilon_2 = \sqrt{\epsilon_1}$. One reason is for more generality that allows us to compare several recent results and another reason is that having different accuracy levels for the first-order and the second-order guarantee brings more flexibility in the choice of our algorithms.

Recently, there has emerged a surge of studies interested in finding an approximate second-order stationary point that satisfy (2). An effective technique used in many algorithms is negative curvature

descent (NCD), which utilizes a negative curvature direction to decrease the objective value. NCD has two additional benefits (i) escaping from non-degenerate saddle points; (ii) searching for a region where the objective function is almost-convex that enables accelerated gradient methods. It has been leveraged to design deterministic and stochastic non-convex optimization with state-of-the-art time complexities for finding a second-order stationary point [4, 17, 16, 2]. A common feature of these algorithms is that they need to compute a negative curvature direction that approximates the eigenvector corresponding to the smallest eigen-value to an accurate level matching the target precision $\epsilon_2$ on the second-order information, i.e., finding a unit vector $\mathbf{v}$ such that $\lambda_{\min}(\nabla^2 f(\mathbf{x})) \geq \mathbf{v}^\top \nabla^2 f(\mathbf{x})\mathbf{v} - \epsilon_2/2$. The approximation accuracy has a direct impact on the complexity of computing the negative curvature. For example, when the Lanczos method is utilized for computing the negative curvature, its complexity (or the number of Hessian-vector products) is in the order of $\widetilde{O}(1/\sqrt{\epsilon_2})$. One potential issue is that the target precision $\epsilon_2$ is usually set to be very small in order to find a high quality solution, which increases the complexity for computing a negative curvature, e.g., the number of Hessian-vector products used in the Lanczos method.

In this paper, we propose an adaptive NCD step based on full or sub-sampled Hessian that uses a noisy negative curvature to update the solution with an error of approximating the smallest eigen-value adaptive to the magnitude of the (stochastic) gradient at the time of invocation. A novel result is that for an iteration $t$ that requires a negative curvature direction it is enough to compute a noisy negative curvature that approximates the smallest eigen-vector with a noise level of $\max(\epsilon_2, \|\mathbf{g}(\mathbf{x}_t)\|^\alpha)$, where $\mathbf{g}(\mathbf{x}_t)$ is the gradient or mini-batch stochastic gradient at the current solution $\mathbf{x}_t$ and $\alpha \in (0, 1]$ is a parameter that characterizes the relationship between $\epsilon_2$ and $\epsilon_1$, i.e., $\epsilon_2 = \epsilon_1^\alpha$. It implies that the Lanczos method only needs $\widetilde{O}(1/\sqrt{\max(\epsilon_2, \|\mathbf{g}(\mathbf{x}_t)\|^\alpha)})$ number of Hessian-vector products for computing such a noisy negative curvature. Another feature of the proposed adaptive NCD step is that it encourages the competition between a negative curvature descent and the gradient descent to guarantee a maximal decrease of the objective value. Building on the proposed adaptive NCD step, we design two simple algorithms to enjoy a second-order convergence for deterministic and stochastic non-convex optimization. Furthermore, we demonstrate the applications of the proposed adaptive NCD steps in existing deterministic and stochastic optimization algorithms to match the state-of-the-art worst-case complexity for finding a second-order stationary point. However, the adaptive nature of the developed algorithms make them perform better than their counterparts using the standard NCD step.

## 2 Related Work

There have been several recent studies that explicitly explore the negative curvature direction for updating the solution. Here, we emphasize the differences between the development in this paper and previous works. Curtis and Robinson [7] proposed a similar algorithm to one of our deterministic algorithms except for how to compute the negative curvature. The key difference between our work and [7] lie at they ignored the computational costs for computing the (approximate) negative curvature. In addition, they considered a stochastic version of their algorithms but provided no second-order convergence guarantee. In contrast, we also develop a stochastic algorithm with provable second-order convergence guarantee.

Royer and Wright [17] proposed an algorithm that utilizes the negative gradient direction, the negative curvature direction, the Newton direction and the regularized Newton direction together with line search in a unified framework, and also analyzed the time complexity of a variant with inexact calculations of the negative curvature by the Lanczos algorithm and of the (regularized) Newton directions by conjugate gradient method. The comparison between their algorithm and our algorithms shows that (i) we only use the gradient and the negative curvature directions; (ii) the time complexity for computing an approximate negative curvature in their work is also of the order of $\widetilde{O}(1/\sqrt{\epsilon_2})$; (iii) the time complexity of one of our deterministic algorithm is at least the same and usually better than their time complexity. Additionally, their conjugate gradient method could fail due to the inexact smallest eigen-value computed by the randomized Lanczos method, and their first-order and second-order convergence guarantee could be on different points.

Carmon et al. [4] developed an algorithm that utilizes the negative curvature descent to reach a region that is almost convex and then switches to an accelerated gradient method to decrease the magnitude of the gradient. One of our algorithms is built on this development by replacing their negative

curvature descent with our adaptive negative curvature descent, which has the same guarantee on the smallest eigen-value of the returned solution but uses a much less number of Hessian-vector products. In addition, we also show that an inexact Hessian can be used in place of the full Hessian to enjoy the same iteration complexity. Several studies revolve around solving cubic regularization step [1, 18], which also requires a negative curvature direction.

Recently, several stochastic algorithms use the negative curvature information to derive the state-of-the-art time complexities for finding a second-order stationary point for non-convex optimization [16, 2, 19, 3], which combine existing stochastic first-order algorithms and a NCD method with differences lying at how to compute the negative curvature. In this work, we also demonstrate the applications of the proposed adaptive NCD for stochastic non-convex optimization, and develop several stochastic algorithms that not only match the state-of-the-art worst-case time complexity but also enjoy adaptively smaller time complexity for computing the negative curvature. We emphasize that the proposed adaptive NCD could be used in future developments of non-convex optimization.

## 3 Preliminaries and Warm-up

In this work, we will consider two types of non-convex optimization problem: deterministic objective where the gradient $\nabla f(\mathbf{x})$ and Hessian $\nabla^2 f(\mathbf{x})$ can be computed, stochastic objective $f(\mathbf{x}) = \mathbb{E}_\xi[f(\mathbf{x}; \xi)]$ where only stochastic gradient $\nabla f(\mathbf{x}; \xi)$ and stochastic Hessian $\nabla^2 f(\mathbf{x}; \xi)$ can be computed. We note that a finite-sum objective can be considered as a stochastic objective. The goal of the paper is to design algorithms that can find an $(\epsilon_1, \epsilon_2)$-second order stationary point $\mathbf{x}$ that satisfies (2). For simplicity, we consider $\epsilon_2 = \epsilon_1^\alpha$ for $\alpha \in (0, 1]$.

A function $f(\mathbf{x})$ is smooth if its gradient is Lipschitz continuous, i.e., there exists $L_1 > 0$ such that $\|\nabla f(\mathbf{x}) - \nabla f(\mathbf{y})\| \leq L_1\|\mathbf{x} - \mathbf{y}\|$ hold for all $\mathbf{x}, \mathbf{y}$. The Hessian of a twice differentiable function $f(\mathbf{x})$ is Lipschitz continuous, if there exists $L_2 > 0$ such that $\|\nabla^2 f(\mathbf{x}) - \nabla^2 f(\mathbf{y})\|_2 \leq L_2\|\mathbf{x} - \mathbf{y}\|$ for all $\mathbf{x}, \mathbf{y}$, where $\|X\|_2$ denotes the spectral norm of a matrix $X$. A function $f(\mathbf{x})$ is called $\mu$-strongly convex ($\mu > 0$) if $f(\mathbf{y}) \geq f(\mathbf{x}) + \nabla f(\mathbf{x})^\top (\mathbf{y} - \mathbf{x}) + \frac{\mu}{2}\|\mathbf{y} - \mathbf{x}\|^2, \forall \mathbf{x}, \mathbf{y}$. If $f(\mathbf{x})$ satisfies the above condition for $\mu < 0$, it is referred to as $\gamma$-almost convex with $\gamma = -\mu$.

Throughout the paper, we make the following assumptions.

**Assumption 1.** *For the optimization problem (1), we assume:*

  *(i) the objective function $f(\mathbf{x})$ is twice differentiable;*

  *(ii) it has $L_1$-Lipschitz continuous gradient and $L_2$-Lipschitz continuous Hessian;*

  *(iii) given an initial solution $\mathbf{x}_0$, there exists $\Delta < \infty$ such that $f(\mathbf{x}_0) - f(\mathbf{x}_*) \leq \Delta$, where $\mathbf{x}_*$ denotes the global minimum of (1);*

  *(iv) if $f(\mathbf{x})$ is a stochastic objective, we assume each random function $f(\mathbf{x}; \xi)$ is twice differentiable and has $L_1$-Lipschitz continuous gradient and $L_2$-Lipschitz continuous Hessian, and its stochastic gradient has exponential tail behavior, i.e., $\mathbb{E}[\exp(\|\nabla f(\mathbf{x}; \xi) - \nabla f(\mathbf{x})\|^2/G^2)] \leq \exp(1)$ holds for any $\mathbf{x} \in \mathbb{R}^d$;*

  *(v) a Hessian-vector product can be computed in $O(d)$ time.*

In this paper, we assume there exists an algorithm that can compute a unit-length negative curvature direction $\mathbf{v} \in \mathbb{R}^d$ of a function $f(\mathbf{x})$ satisfying

$$\lambda_{\min}(\nabla^2 f(\mathbf{x})) \geq \mathbf{v}^\top \nabla^2 f(\mathbf{x})\mathbf{v} - \varepsilon \tag{3}$$

with high probability $1 - \delta$. We refer to such an algorithm as $\text{NCS}(f, \mathbf{x}, \varepsilon, \delta)$ and denote its time complexity by $T_n(f, \varepsilon, \delta, d)$, where NCS is short for negative curvature search.

There exist algorithms to implement negative curvature search (NCS) for two different cases: deterministic objective and stochastic objective with theoretical guarantee, which we provide in the supplement. To facilitate the discussion in the following sections, we summarize the results here.

**Lemma 1.** *For a deterministic objective, the Lanczos method find a unit vector $\mathbf{v}$ satisfying (3) with a time complexity of $T_n(f, \varepsilon, \delta, d) = \widetilde{O}\left(\frac{d}{\sqrt{\varepsilon}}\right)$. For a stochastic objective $f(\mathbf{x}) = \mathbb{E}_\xi[f(\mathbf{x}; \xi)]$, there exists a randomized algorithm that produces a unit vector $\mathbf{v}$ satisfying (3) with a time complexity*

---

**Algorithm 1** $\text{AdaNCD}^{\text{det}}(\mathbf{x}, \alpha, \delta, \nabla f(\mathbf{x}))$

---

1: Apply $\text{NCS}(f, \mathbf{x}, \frac{\max(\epsilon_2, \|\nabla f(\mathbf{x})\|^\alpha)}{2}, \delta)$ to find a unit vector $\mathbf{v}$ satisfying (3)
2: **if** $\frac{2(-\mathbf{v}^\top \nabla^2 f(\mathbf{x})\mathbf{v})^3}{3L_2^2} > \frac{\|\nabla f(\mathbf{x})\|^2}{2L_1}$ **then**
3:     $\mathbf{x}^+ = \mathbf{x} - \frac{2|\mathbf{v}^\top \nabla^2 f(\mathbf{x})\mathbf{v}|}{L_2}\text{sign}(\mathbf{v}^\top \nabla f(\mathbf{x}))\mathbf{v}$
4: **else**
5:     $\mathbf{x}^+ = \mathbf{x} - \frac{1}{L_1}\nabla f(\mathbf{x})$
6: **end if**
7: **Return** $\mathbf{x}^+, \mathbf{v}$

---

**Algorithm 2** $\text{AdaNCD}^{\text{mb}}(\mathbf{x}, \alpha, \delta, \mathcal{S}, \mathbf{g}(\mathbf{x}))$:

---

1: Apply $\text{NCS}(f_{\mathcal{S}}, \mathbf{x}, \frac{\max(\epsilon_2, \|\mathbf{g}(\mathbf{x})\|^\alpha)}{2}, \delta)$ to find a unit vector $\mathbf{v}$ satisfying (3) for $f_{\mathcal{S}}$
2: **if** $\frac{2(-\mathbf{v}^\top H_{\mathcal{S}}(\mathbf{x})\mathbf{v})^3}{3L_2^2} - \frac{\epsilon_2 |\mathbf{v}^\top H_{\mathcal{S}}(\mathbf{x})\mathbf{v}|^2}{6L_2^2} > \frac{\|\mathbf{g}(\mathbf{x})\|^2}{4L_1} - \frac{\epsilon'^2}{L_1}$ **then**
3:     $\mathbf{x}^+ = \mathbf{x} - \frac{2|\mathbf{v}^\top H_{\mathcal{S}}(\mathbf{x})\mathbf{v}|}{L_2}z\mathbf{v}$         $\diamond z \in \{1, -1\}$ is a Rademacher random variable
4: **else**
5:     $\mathbf{x}^+ = \mathbf{x} - \frac{1}{L_1}\mathbf{g}(\mathbf{x})$
6: **end if**
7: **Return** $\mathbf{x}^+, \mathbf{v}$

---

*of with $T_n(f, \varepsilon, \delta, d) = \widetilde{O}\left(\frac{d}{\varepsilon^2}\right)$. If $f(\mathbf{x})$ has a finite-sum structure with $m$ components, then a randomized algorithm exists that produces a unit vector $\mathbf{v}$ satisfying (3) with a time complexity of $T_n(f, \varepsilon, \delta, d) = \widetilde{O}(d(m + m^{3/4}\sqrt{1/\varepsilon}))$, where $\widetilde{O}$ suppresses a logarithmic term in $\delta, d, 1/\varepsilon$.*

## 4 Adaptive Negative Curvature Descent Step

In this section, we present several variants of adaptive negative curvature descent (AdaNCD) step for different objectives and with different available information. We also present their guarantee on decreasing the objective function.

### 4.1 Deterministic Objective

For a deterministic objective, when a negative curvature of the Hessian matrix $\nabla^2 f(\mathbf{x})$ at a point $\mathbf{x}$ is required, the gradient $\nabla f(\mathbf{x})$ is readily available. We utilize this information to design an AdaNCD shown in Algorithm 1. First, we compute a noisy negative curvature $\mathbf{v}$ that approximates the smallest eigen-value of the Hessian at the current point $\mathbf{x}$ up to a noise level $\varepsilon = \max(\epsilon_2, \|\nabla f(\mathbf{x})\|^\alpha)$. Then we take either the noisy negative curvature direction or the negative gradient direction depending on which decreases the objective value more. This is done by comparing the estimations of the objective decrease for following these two directions as shown in Step 3 in Algorithm 1. Its guarantee on objective decrease is stated in the following lemma, which will be useful for proving convergence to a second-order stationary point.

**Lemma 2.** *When $\mathbf{v}^\top \nabla^2 f(\mathbf{x})\mathbf{v} \leq 0$, the Algorithm 1 (AdaNCD$^{det}$) provides a guarantee that*

$$f(\mathbf{x}) - f(\mathbf{x}^+) \geq \max\left(\frac{2|\mathbf{v}^\top \nabla^2 f(\mathbf{x})\mathbf{v}|^3}{3L_2^2}, \frac{\|\nabla f(\mathbf{x})\|^2}{2L_1}\right)$$

### 4.2 Stochastic Objective

For a stochastic objective $f(\mathbf{x}) = \mathbb{E}_\xi[f(\mathbf{x}; \xi)]$, we assume a noisy gradient $\mathbf{g}(\mathbf{x})$ that satisfies (4) (with high probability) is available when computing the negative curvature at $\mathbf{x}$:

$$\|\mathbf{g}(\mathbf{x}) - \nabla f(\mathbf{x})\| \leq \epsilon' \tag{4}$$

This can be met by using a mini-batch stochastic gradient $\mathbf{g}(\mathbf{x}) = \frac{1}{|\mathcal{S}_1|}\sum_{\xi \in \mathcal{S}_1} f(\mathbf{x}; \xi)$ with a sufficiently large batch size (see Lemma 9 in the supplement).

We can use a NCS algorithm to compute a negative curvature based on a mini-batched Hessian. To this end, let $H_{\mathcal{S}}(\mathbf{x}) = \frac{1}{|\mathcal{S}|}\sum_{\xi \in \mathcal{S}} \nabla^2 f(\mathbf{x}; \xi)$, where $\mathcal{S}$ denote a set of random samples, satisfy the

following inequality (with high probability):

$$\|H_{\mathcal{S}}(\mathbf{x}) - \nabla^2 f(\mathbf{x})\|_2 \leq \epsilon_2/12. \tag{5}$$

The inequality (5) holds with high probability when $\mathcal{S}$ is sufficiently large due to the exponential tail behavior of $\|H_{\mathcal{S}}(\mathbf{x}) - \nabla^2 f(\mathbf{x})\|_2$ stated in the Lemma 8 in the supplement.

Denote by $f_{\mathcal{S}} = \frac{1}{|\mathcal{S}|} \sum_{\xi \in \mathcal{S}} f(\cdot; \xi)$. A variant of AdaNCD using such a mini-batched Hessian is presented in Algorithm 2, where $z$ is a Rademacher random variable, i.e. $z = 1, -1$ with equal probability. Lemma 3 provides objective decrease guarantee of Algorithm 2.

**Lemma 3.** *When $\mathbf{v}^\top H_{\mathcal{S}}(\mathbf{x})\mathbf{v} \leq 0$ and (5) holds (with high probability), the Algorithm 2 (AdaNCD$^{mb}$) provides a guarantee (with high probability) that*

$$f(\mathbf{x}) - \mathbb{E}[f(\mathbf{x}^+)] \geq \max\left\{ \frac{2(-\mathbf{v}^\top H_{\mathcal{S}}(\mathbf{x})\mathbf{v})^3}{3L_2^2} - \frac{\epsilon_2 |\mathbf{v}^\top H_{\mathcal{S}}(\mathbf{x})\mathbf{v}|^2}{6L_2^2}, \frac{\|\mathbf{g}(\mathbf{x})\|^2}{4L_1} - \frac{\epsilon'^2}{L_1} \right\}$$

*If $\mathbf{v}^\top H_{\mathcal{S}}(\mathbf{x})\mathbf{v} \leq -\epsilon_2/2$, we have*

$$f(\mathbf{x}) - f(\mathbf{x}^+) \geq \max\left( \frac{\epsilon_2^3}{24L_2^2}, \frac{\|\mathbf{g}(\mathbf{x})\|^2}{4L_1} - \frac{\epsilon'^2}{L_1} \right)$$

**Remark:** When the objective has a finite-sum structure, Algorithm 2 is also applicable, where the noise gradient $\mathbf{g}(\mathbf{x})$ can be replaced with the full gradient $\nabla f(\mathbf{x})$. This is the variant using sub-sampled Hessian.

We can also use a different variant of Algorithm 2: AdaNCD$^{online}$, which uses an online algorithm to compute the negative curvature and is described in Algorithm 7 (in the supplement) with Lemma 7 (in the supplement) as its theoretical guarantee.

## 5 Simple Adaptive Algorithms with Second-order Convergence

In this section, we present simple deterministic and stochastic algorithms by employing AdaNCD presented in the last section. These simple algorithms deserve attention due to several reasons (i) they are simpler than many previous algorithms but can enjoy a similar time complexity when $\epsilon_2 = \epsilon_1$; (ii) they guarantee that the objective value can decrease at every iteration, which does not hold for some complicated algorithms with state-of-the-art complexity results (e.g., [4, 2]).

### 5.1 Deterministic Objective

We present a deterministic algorithm for a deterministic objective in Algorithm 3, which is referred to as AdaNCG (where NCG represents Negative Curvature and Gradient, Ada represents the adaptive nature of the NCD component).

**Theorem 1.** *For any $\alpha \in (0, 1]$, the AdaNCG algorithm terminates at iteration $j_*$ for some*

$$j_* \leq 1 + \max\left( \frac{12L_2^2}{\epsilon_1^{3\alpha}}, \frac{2L_1}{\epsilon_1^2} \right)(f(\mathbf{x}_1) - f(\mathbf{x}_{j_*})) \leq 1 + \max\left( \frac{12L_2^2}{\epsilon_1^{3\alpha}}, \frac{2L_1}{\epsilon_1^2} \right)\Delta, \tag{6}$$

*with $\|\nabla f(\mathbf{x}_{j_*})\| \leq \epsilon_1$, and with probability at least $1 - \delta$, $\lambda_{\min}(\nabla^2 f(\mathbf{x}_{j_*})) \geq -\epsilon_1^\alpha$. Furthermore, the $j$-th iteration requires time a complexity of $T_n(f, \max(\epsilon_1^\alpha, \|\nabla f(\mathbf{x}_j)\|^\alpha), \delta', d)$.*

**Remark:** First, when $\epsilon_2 = \epsilon_1 = \epsilon$, the iteration complexity of AdaNCG for achieving a point with $\max\{\|\nabla f(\mathbf{x})\|, -\lambda_{\min}(\nabla^2 f(\mathbf{x}))\} \leq \epsilon$ is $O(1/\epsilon^3)$, which match the results in previous works (e.g. [17, 5, 6, 18]). However, the number of Hessian-vector products in AdaNCG could be much less than that in these existing works. For example, the number of Hessian-vector products in [17, 18] is $\widetilde{O}(1/\sqrt{\epsilon_2})$ at each iteration requiring the second-order information. In contrast, when employing the Lanczos method the number of Hessian-vector products at each iteration of AdaNCG is $\widetilde{O}(d/\sqrt{\max(\epsilon_2, \|\nabla f(\mathbf{x}_j)\|^\alpha)})$, which could be much smaller than $\widetilde{O}(1/\sqrt{\epsilon_2})$ depending on the magnitude of the gradient. Second, the worse-case time complexity of AdaNCG is given by $\widetilde{O}\left(d \max\left\{ \epsilon_1^{-2}\epsilon_2^{-1/2}, \epsilon_2^{-7/2} \right\}\right)$ using the worse-case time complexity of each iteration, which is the same as the result of Theorem 2 in [18].

One might notice that if we plug $\epsilon_1 = \epsilon$, $\epsilon_2 = \sqrt{\epsilon}$ into the worst-case time complexity of AdaNCG, we end up with $\widetilde{O}(d/\epsilon^{9/4})$, which is worse than the best time complexity $\widetilde{O}(d/\epsilon^{7/4})$ found in literature

**Algorithm 3** AdaNCG: $(\mathbf{x}_0, \epsilon_1, \alpha, \delta)$

---

1: $\mathbf{x}_1 = \mathbf{x}_0$, $\epsilon_2 = \epsilon_1^{\alpha}$
2: $\delta' = \delta / (1 + \max\left(\frac{12L_2^2}{\epsilon_2^3}, \frac{2L_1}{\epsilon_1^2}\right)\Delta)$,
3: **for** $j = 1, 2, \ldots$, **do**
4:     $(\mathbf{x}_{j+1}, \mathbf{v}_j) = \text{AdaNCD}^{\text{det}}(\mathbf{x}_j, \alpha, \delta', \nabla f(\mathbf{x}))$
5:     **if** $\mathbf{v}_j^\top \nabla^2 f(\mathbf{x}_j)\mathbf{v}_j > -\frac{\epsilon_2}{2}$ and $\|\nabla f(\mathbf{x}_j)\| \leq \epsilon_1$ **then**
6:         **Return** $\mathbf{x}_j$
7:     **end if**
8: **end for**

---

**Algorithm 4** S-AdaNCG: $(\mathbf{x}_0, \epsilon_1, \alpha, \delta)$

---

1: $\mathbf{x}_1 = \mathbf{x}_0$, $\epsilon_2 = \epsilon_1^{\alpha}$, $\delta' = \delta/\widetilde{O}(\epsilon_1^{-2}, \epsilon_2^{-3})$
2: **for** $j = 1, 2, \ldots$, **do**
3:     Generate two random sets $\mathcal{S}_1, \mathcal{S}_2$
4:     let $\mathbf{g}(\mathbf{x}_j) = \frac{1}{|\mathcal{S}_1|}\sum_{\xi \in \mathcal{S}_1} \nabla f(\mathbf{x}; \xi)$ satisfy (4)
5:     $(\mathbf{x}_{j+1}, \mathbf{v}_j) = \text{AdaNCD}^{\text{mb}}(\mathbf{x}_j, \alpha, \delta', \mathcal{S}_2, \mathbf{g}(\mathbf{x}_j))$
6:     **if** $\mathbf{v}_j^\top H_{\mathcal{S}_2}(\mathbf{x}_j)\mathbf{v}_j > -\epsilon_2/2$ and $\|\mathbf{g}(\mathbf{x}_j)\| \leq \epsilon_1$ **then**
7:         **Return** $\mathbf{x}_j$
8:     **end if**
9: **end for**

---

(e.g., [1, 4]). In next section, we will use AdaNCG as a sub-routine to develop an algorithm that can match the state-of-the-art time complexity but also enjoy the adaptiveness of AdaNCG.

Before ending this subsection, we would like to point out that an inexact Hessian satisfying (5) can be used for computing the negative curvature. For example, if the objective has a finite-sum form, AdaNCD$^{\text{det}}$ can be replaced by AdaNCD$^{\text{mb}}$ using full gradient. Lemma 3 provides a similar guarantee to Lemma 2 and can be used to derive a similar convergence to Theorem 1.

## 5.2 Stochastic Objective

We present a stochastic algorithm based on the AdaNCD$^{\text{mb}}$ in Algorithm 4, which is referred to as S-AdaNCG (where S represents stochastic). A similar algorithm based on the AdaNCD$^{\text{online}}$ with similar worst-case complexity can be developed, which is omitted.

**Theorem 2.** *Set* $|\mathcal{S}_1| = \frac{32G^2}{\epsilon_1^2}(1 + 3\log(\frac{2}{\delta'}))$ *and* $|\mathcal{S}_2| = \frac{9216L_1^2}{\epsilon_2^2}\log(\frac{4d}{\delta'})$. *With probability* $1 - \delta$, *the S-AdaNCG algorithm terminates at some iteration* $j_* = \widetilde{O}(\max\left(\frac{1}{\epsilon_2^3}, \frac{1}{\epsilon_1^2}\right))$ *and upon termination it holds that* $\|\nabla f(\mathbf{x}_{j_*})\| \leq 2\epsilon_1$ *and* $\lambda_{min}\left(\nabla^2 f(\mathbf{x}_{j_*})\right) \geq -2\epsilon_2$ *with probability* $1 - 3\delta$. *Furthermore, the worst-case time complexity of S-AdaNCG is given by* $\widetilde{O}\left(\max\left(\frac{1}{\epsilon_2^3}, \frac{1}{\epsilon_1^2}\right)\left(\frac{d}{\epsilon_1^2} + T_n(f_{\mathcal{S}_2}, \epsilon_2, \delta', d)\right)\right)$.

**Remark:** We can analyze the worst-case time complexity of S-AdaNCG by using randomized algorithms as in Lemma 1 to compute the negative curvature with $T_n(f, \varepsilon, \delta, d) = \widetilde{O}\left(\frac{d}{\varepsilon^2}\right)$. Let us consider $\epsilon_2 = \epsilon_1^{1/2}$, it is not difficult to show that the worst-case time complexity of S-AdaNCG is $\widetilde{O}\left(d/\epsilon_1^4\right)$, which matches the time complexity of stochastic gradient descent for finding a first-order stationary point. It is almost linear in the problem's dimensionality better than that of noisy SGD methods [9, 20]. It is also notable that the worst-case time complexity of S-AdaNCG is worse than that of a recent algorithm called Natasha2 proposed in [2], which has a state-of-the-art time complexity of $\widetilde{O}\left(d/\epsilon_1^{3.5}\right)$ for finding an $(\epsilon_1, \sqrt{\epsilon_1})$ second-order stationary point. However, S-AdaNCG is much simpler than Natasha2, which involves many parameters and switches between several procedures. In next section, we will present an improved algorithm of S-AdaNCG, whose worst-case time complexity matches $\widetilde{O}\left(d/\epsilon_1^{3.5}\right)$ for finding an $(\epsilon_1, \sqrt{\epsilon_1})$ second-order stationary point.

## 6 Adaptive Algorithms with State-of-the-Art Complexities

In this section, we demonstrate the applications of the presented AdaNCD for deterministic and stochastic optimization with a state-of-the-art time complexity, aiming for better practical performance than their counterparts in literature. We will show that how the proposed AdaNCD can reduce the time complexity of these existing algorithms.

**Algorithm 5** AdaNCG$^+$: $(\mathbf{x}_0, \epsilon_1, \alpha, \delta)$

---

1: $\epsilon_2 = \epsilon_1^\alpha$, $K := \lceil 1 + \Delta \left( \frac{\max(12L_2^2, 2L_1)}{\epsilon_2^3} + \frac{2\sqrt{10}L_2}{\epsilon_1 \epsilon_2} \right) \rceil$
2: $\delta' := \delta/K$
3: **for** $k = 1, 2, \ldots,$ **do**
4:      $\widehat{\mathbf{x}}_k = \text{AdaNCG}(\mathbf{x}_k, \epsilon_1^{3\alpha/2}, \frac{2}{3}, \delta')$
5:      **if** $\|\nabla f(\widehat{\mathbf{x}}_k)\| \le \epsilon_1$ **then**
6:         **Return** $\widehat{\mathbf{x}}_k$
7:      **else**
8:         $f_k(\mathbf{x}) = f(\mathbf{x}) + L_1 \left( [\|\mathbf{x} - \widehat{\mathbf{x}}_k\| - \epsilon_2/L_2]_+ \right)^2$
9:         $\mathbf{x}_{k+1} = \text{Almost-Cvx-AGD}(f_j, \widehat{\mathbf{x}}_k, \frac{\epsilon_1}{2}, 3\epsilon_2, 5L_1)$
10:      **end if**
11: **end for**

---

**Algorithm 6** AdaNCD-SCSG: $(\mathbf{x}_0, \epsilon_1, \alpha, b, \delta)$

---

1: **Input**: $\mathbf{x}_0, \epsilon_1, \alpha, \delta$
2: **for** $j = 1, 2, \ldots,$ **do**
3:      Generate three random sets $\mathcal{S}, \mathcal{S}_1, \mathcal{S}_2$
4:      $\mathbf{y}_j = \text{SCSG-Epoch}(\mathbf{x}_j, \mathcal{S}, b)$
5:      let $\mathbf{g}(\mathbf{y}_j) = \nabla f_{\mathcal{S}_1}(\mathbf{x}; \xi)$ satisfy (4)
6:      $(\mathbf{x}_{j+1}, \mathbf{v}_j) = \text{AdaNCD}^{\text{mb}}(\mathbf{y}_j, \alpha, \delta, \mathcal{S}_2, \mathbf{g}(\mathbf{y}_j))$
7:      **if** $\mathbf{v}_j^\top H_{\mathcal{S}_2}(\mathbf{y}_j)\mathbf{v}_j > -\epsilon_2/2$ and $\|\mathbf{g}(\mathbf{y}_j)\| \le \epsilon_1$ **then**
8:         **Return** $\mathbf{y}_j$
9:      **end if**
10: **end for**

---

## 6.1 Deterministic Objective

For deterministic objective, we consider the accelerated method proposed in [4], which relies on NCD to find a point around which the objective function is almost convex and then switches to an accelerated gradient method. We present an adaptive variant of [4]'s method in Algorithm 5, where we use our AdaNCG in place of NCD. The procedure Almost-Convex-AGD is the same as in [4]. For completeness, we present it in the supplement. The convergence guarantee is presented below.

**Theorem 3.** *For any* $\alpha \in (0, 1]$, *let* $\epsilon_2 = \epsilon_1^\alpha$. *With probability at least* $1 - \delta$, *the Algorithm AdaNCG$^+$ returns a vector* $\widehat{\mathbf{x}}_k$ *such that* $\|\nabla f(\widehat{\mathbf{x}}_k)\| \le \epsilon_1$ *and* $\lambda_{min}(\nabla^2 f(\widehat{\mathbf{x}}_k)) \ge -\epsilon_2$ *with at most* $O\left( \frac{1}{\epsilon_2^3} + \frac{1}{\epsilon_1 \epsilon_2} \right)$ *AdaNCD steps in AdaNCG and* $\widetilde{O}\left[ \left( \frac{1}{\epsilon_2^{7/2}} + \frac{1}{\epsilon_1 \epsilon_2^{3/2}} \right) + \frac{\epsilon_2^{1/2}}{\epsilon_1^2} \right]$ *gradient steps in Almost-Convex-AGD, and each step* $j$ *within AdaNCG$^+$ requires time of* $T_n(f, \max(\epsilon_2, \|\nabla f(\mathbf{x}_j)\|^{2/3})^{1/2}, \delta', d)$, *and the worse-case time complexity of AdaNCG$^+$ is* $\widetilde{O}\left( \left( \frac{d}{\epsilon_1 \epsilon_2^{3/2}} + \frac{d}{\epsilon_2^{7/2}} \right) + \frac{d\epsilon_2^{1/2}}{\epsilon_1^2} \right)$ *when using the Lanczos method for NCS.*

**Remark:** First, when $\epsilon_2 \le \sqrt{\epsilon_1}$, the worst-case time complexity of AdaNCG$^+$ is $\widetilde{O}\left( \frac{d}{\epsilon_1 \epsilon_2^{3/2}} + \frac{d}{\epsilon_2^{7/2}} \right)$. Specially, for $\epsilon_2 = \sqrt{\epsilon_1}$ it reduces to $\widetilde{O}(d/\epsilon^{7/4})$, which matches the best time complexity in previous studies. Second, we note that the subroutine AdaNCG$(\mathbf{x}_j, \epsilon_1^{3\alpha/2}, 2/3, \delta')$ provides the same guarantee as the NCD in [4] (see Corollary 1 in the supplement), i.e., returning a solution $\widehat{\mathbf{x}}_j$ satisfying $\lambda_{\min}(\nabla^2 f(\widehat{\mathbf{x}}_j)) \ge -\epsilon_2$ with high probability. The number of iterations within AdaNCG is similar to that in NCD employed by [4], and the number of iterations within Almost-Convex-AGD is similar to that in [4]. The improvement of AdaNCG$^+$ over [4]'s algorithm is brought by reducing the number of Hessian-vector products for performing each iteration of AdaNCG. In particular, the number of Hessian-vector products of each NCD step in [4] is $\widetilde{O}(1/\sqrt{\epsilon_2})$, which becomes $\widetilde{O}(1/\sqrt{\max(\epsilon_2, \|\nabla f(\mathbf{x}_j)\|^{2/3})})$ for each AdaNCD step in AdaNCG$^+$. Finally, we note that AdaNCG$^+$ has the same worse-case time complexity as AdaNCG for $\epsilon_2 \in [\epsilon_1, \epsilon_1^{2/3}]$, but improves over AdaNCG$^+$ for $\epsilon_2 \in [\epsilon_1^{2/3}, \epsilon_1^{1/2}]$.

## 6.2 Stochastic Objective

Next, we present a stochastic algorithm for tackling a stochastic objective $f(\mathbf{x}) = \mathbb{E}[f(\mathbf{x}; \xi)]$ in order to achieve a state-of-the-art worse-case complexity for finding a second-order stationary point. We consider combining the proposed AdaNCD$^{\text{mb}}$ with an existing stochastic variance reduced gradient method for a stochastic objective, namely SCSG [12].

The detailed steps are shown in Algorithm 6, which is referred to as AdaNCD-SCSG and can be considered as an improvement of Algorithm 4. The sub-routine SCSG-Epoch is one epoch of SCSG, which is included in the supplement. It is worth mentioning that Algorithm 6 is based on the design of [19] that also combined a NCD step with SCSG to prove the second-order convergence. The difference from [19] is that they studied how to use a first-order method without resorting to Hessian-vector products to extract the negative curvature direction, while we focus on reducing the time complexity of NCS using the proposed adaptive NCD. Our result below shows AdaNCD-SCSG has a worst-case time complexity that matches the state-of-the-art time complexity for finding an $(\epsilon_1, \sqrt{\epsilon_1})$ second-order stationary point.

**Theorem 4.** *For any $\alpha \in (0, 1]$, let $\epsilon_2 = \epsilon_1^\alpha$. Suppose $|\mathcal{S}| = \widetilde{O}(\max(1/\epsilon_1^2, 1/(\epsilon_2^{9/2}b^{1/2})))$, $|\mathcal{S}_1| = \widetilde{O}(1/\epsilon_1^2)$ and $|\mathcal{S}_2| = \widetilde{O}(1/\epsilon_2^2)$. With high probability, the Algorithm AdaNCD-SCSG returns a vector $\mathbf{y}_j$ such that $\|\nabla f(\mathbf{y}_j)\| \leq 2\epsilon_1$ and $\lambda_{min}(\nabla^2 f(\mathbf{x}_j)) \geq -2\epsilon_2$ with at most $\widetilde{O}\left( \frac{b^{1/3}}{\epsilon_1^{4/3}} + \frac{1}{\epsilon_2^3} \right)$ calls of SCSG-Epoch and AdaNCD$^{\text{mb}}$.*

**Remark:** The worst-case time complexity of AdaNCD-SCSG can be computed as

$$\widetilde{O}\left( \left( \frac{b^{1/3}}{\epsilon_1^{4/3}} + \frac{1}{\epsilon_2^3} \right) (|\mathcal{S}|d + |\mathcal{S}_1|d + T_n(f_{\mathcal{S}_2}, \epsilon_2, \delta', d)) \right).$$

If we consider using randomized algorithms as in Lemma 6 in supplement to implement NCS in AdaNCD$^{\text{mb}}$, the above time complexity reduces to $\widetilde{O}\left( d \left( \frac{b^{1/3}}{\epsilon_1^{4/3}} + \frac{1}{\epsilon_2^3} \right) \left( \frac{1}{\epsilon_1^2} + \frac{1}{\epsilon_2^{9/2}b^{1/2}} + \frac{1}{\epsilon_2^2} \right) \right)$. Let us consider $\epsilon_2 = \epsilon_1^{1/2}$. By setting $b = 1/\epsilon_1^{1/2}$, the worst-case time complexity of AdaNCD-SCSG is $\widetilde{O}(d/\epsilon^{3.5})$.

## 7 Empirical Studies

In this section, we report some experimental results to justify effectiveness of AdaNCD for both deterministic and stochastic non-convex optimization. We consider three problems, namely, the cubic regularization, regularized non-linear least-square, and one hidden-layer neural network (NN) problem.

The cubic regularization problem is: $\min_\mathbf{w} \frac{1}{2}\mathbf{w}^\top A\mathbf{w} + \mathbf{b}^\top \mathbf{w} + \frac{\rho}{3}\|\mathbf{w}\|_2^3$, where $A \in \mathbb{R}^{1000 \times 1000}$. For deterministic optimization, we generate a diagonal $A$ such that 100 randomly selected diagonal entry is $-1$ and the rest diagonal entries follow uniform distribution between $[1, 2]$, and set $\mathbf{b}$ as a zero vector. For stochastic optimization, we let $A = A' + \mathbb{E}[\text{diag}(\xi)]$ and $\mathbf{b} = \mathbb{E}[\xi']$, where $A'$ is generated similarly, $\xi$ are uniform random variables from $[-0.1, 0.1]$ and $\xi'$ are uniform random variables from $[-1, 1]$. The parameter $\rho$ is set to $0.5$ for both deterministic and stochastic experiments. It is clear that zero is a saddle point of the problem. In order to test the capability of escaping from saddle point, we let each algorithm start from a zero vector.

The regularized non-linear least-square problem is: $\min_\mathbf{w} \frac{1}{n}\sum_{i=1}^n \left( y_i - \sigma(\mathbf{w}^\top \mathbf{x}_i) \right)^2 + \sum_{i=1}^d \frac{\lambda w_i^2}{1+\alpha w_i^2}$, where $\mathbf{x}_i \in \mathbb{R}^d$, $y_i \in \{0, 1\}$, $\sigma(s) = 1/(1 + \exp(-s))$ is a sigmoid function, and the second term is a non-convex regularizer [15], which is to increase the negative curvature of the problem. We use w1a data ($n = 2477, d = 300$) from the libsvm website [8], and set $\lambda = 1$.

Learning a NN with one hidden layer is imposed as: $\min_\mathbf{w} \frac{1}{n}\sum_{i=1}^n \ell(W_2\sigma(W_1\mathbf{x}_i + b_1) + b_2, y_i)$, where $\mathbf{x}_i \in \mathbb{R}^d$, $y_i \in \{1, -1\}$ are input data, $W_1, W_2, b_1, b_2$ are parameters of the NN with appropriate dimensions, and $\ell(z, y)$ is cross-entropy loss. We use $12,665$ examples from the MNIST dataset [11] that belong to two categories 0 and 1 as input data, where the input feature dimensionality is 784. The number of neurons in hidden layer is set to 10 so that the total number of parameters including bias terms is 7872.

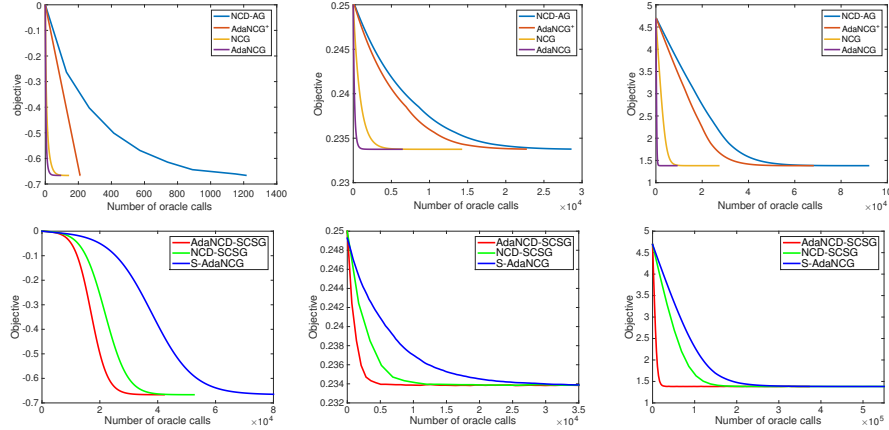

Figure 1: Comparison of different deterministic algorithms (upper) and stochastic algorithms (lower) for solving cubic regularization, regularized nonlinear least square, and neural network (from left to right).

For deterministic experiments, we compare AdaNCG, AdaNCG$^+$ with their non-adaptive counterparts. In particular, the non-adaptive counterpart of AdaNCG named NCG uses NCS$(f, \mathbf{x}, \epsilon_2/2, \delta)$. The non-adaptive counterpart of AdaNCG$^+$ is the algorithm proposed in [4], which is referred to as NCD-AG. For stochastic experiments, we compare S-AdaNCG, AdaNCD-SCSG, and the non-adaptive version of AdaNCD-SCSG named NCD-SCSG. For all experiments, we choose $\alpha = \frac{1}{2}$, i.e., $\epsilon_2 = \sqrt{\epsilon_1}$. The parameters $L_1, L_2$ are tuned for the non-adaptive algorithm NCG, and the same values are used in other algorithms. The searching range for $L_1$ and $L_2$ are from $10^{-5:1:5}$. The mini-batch size used in S-AdaNCG and AdaNCD-SCSG is set to 50 for cubic regularization and 128 for other two tasks. We use the Lanczos method for NCS. For non-adaptive algorithms, the number of iterations in each call of the Lanczos method is set to $\min(C \log(d)/\sqrt{\epsilon_2}, d)$; and for adaptive algorithms, the number of iterations in each call of the Lanczos method is set to $\min(C \log(d)/\sqrt{\max(\epsilon_2, \|\mathbf{g}(\mathbf{x})\|^{1/2})}, d)$, where $\mathbf{g}(\mathbf{x})$ is either a full gradient or a mini-batch stochastic gradient. The value of $C$ is set to $\sqrt{L_1}$.

We set $\epsilon_1 = 10^{-2}$ for cubic regularization, and $\epsilon_1 = 10^{-4}$ for other two tasks. We report the objective value v.s. the number of oracle calls (including gradient evaluations and Hessian-vector productions) in Figure 1. From deterministic optimization results, we can see that the AdaNCD can greatly improve the convergence of AdaNCG and AdaNCG$^+$ compared to their non-adaptive counterparts. In addition, AdaNCG performs better than AdaNCG$^+$ on the tested tasks. The reason is that AdaNCG can guarantee the decrease of the objective values at every iteration, while AdaNCG$^+$ that uses the AG method to optimize an almost convex functions does not have such guarantee. From stochastic optimization results, AdaNCD also makes AdaNCD-SCSG converge faster than its non-adaptive counterpart NCD-SCSG. In addition, AdaNCD-SCSG is faster than S-AdaNCG. Finally, we note that the final solution found by the proposed algorithms satisfy the prescribed optimality condition. For example, on the solution found by AdaNCG on the cubic regularization problem the gradient norm is $0.0085$ and the minimum eigen-value of the Hessian is $-0.0043$.

## 8 Conclusion

In this paper, we have developed several variants of adaptive negative curvature descent step that employ a noisy negative curvature direction for non-convex optimization. The novelty of the proposed algorithms lie at that the noise level in approximating the negative curvature is adaptive to the magnitude of the current gradient instead of a prescribed small noise level, which could dramatically reduce the number of Hessian-vector products. Building on the adaptive negative curvature descent step, we have developed several deterministic and stochastic algorithms and established their complexities. The effectiveness of adaptive negative curvature descent is also demonstrated by empirical studies.

## Acknowledgments

We thank the anonymous reviewers for their helpful comments. M. Liu, T. Yang are partially supported by National Science Foundation (IIS-1545995).

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
