[Supplementary Material · NCG_supplement.pdf]

# Supplementary Material for "Adaptive Negative Curvature Descent with Applications in Non-convex Optimization"

**Mingrui Liu[†], Zhe Li[†], Xiaoyu Wang[‡], Jinfeng Yi[♮], Tianbao Yang[†]**
[†]Department of Computer Science, The University of Iowa, Iowa City, IA 52242, USA
[‡] Intellifusion [♮] JD AI Research
`mingrui-liu, tianbao-yang@uiowa.edu`

## 1 Negative Curvature Search (NCS) for Two Cases

In this section, we introduce specific implementations of Negative Curvature Search (NCS) for two different settings, i.e. deterministic objective and stochastic objective.

**Deterministic Objective** In particular, we have the following lemma.

**Lemma 4.** *Suppose that the Lanczos method is applied to find the largest eigenvalue of $L_1 I - \nabla^2 f(\mathbf{x})$ starting at a random vector uniformly distributed over the unit sphere. Then, for any $\varepsilon > 0$ and $\delta \in (0,1)$, there is a probability at least $1 - \delta$ that the method outputs a unit vector $\mathbf{v}$ such that $\lambda_{\min}(\nabla^2 f(\mathbf{x})) \geq \mathbf{v}^\top \nabla^2 f(\mathbf{x}) \mathbf{v} - \varepsilon$ with at most $\min\left(d, \frac{\log(d/\delta^2)\sqrt{L_1}}{2\sqrt{2\varepsilon}}\right)$ Hessian-vector products. Therefore $T_n(f, \varepsilon, \delta, d) = \widetilde{O}\left(\frac{d}{\sqrt{\varepsilon}}\right)$ provided that $d$ is large enough, , where $\widetilde{O}$ suppresses a logarithmic term in $\delta, d, 1/\varepsilon$.*

**Remark:** The above result follows previous convergence analysis of the Lanczos method [1]. Please refer to [2][Lemma 11] for a proof.

**Stochastic Objective** For a stochastic objective $f(\mathbf{x}) = \mathbb{E}[f(\mathbf{x}; \xi)]$ depending a random variable $\xi$. We can apply Oja's algorithm [3] that iteratively computes $\mathbf{v}_\tau$ by

$$\mathbf{v}_\tau = \frac{(I + \eta \nabla^2 f(\mathbf{x}; \xi_\tau))\mathbf{v}_{\tau-1}}{\|(I + \eta \nabla^2 f(\mathbf{x}; \xi_\tau))\mathbf{v}_{\tau-1}\|} \tag{7}$$

where $\eta$ is a proper step size. The following result provides a guarantee of (3) for an algorithm based on Oja's algorithm.

**Lemma 5.** *Given $\delta \in (0,1)$, there exists an algorithm that generates a solution satisfying (3) with $T_n(\varepsilon, \delta, d) = O\left(\frac{d \log^2(d/\delta)}{\varepsilon^2}\right)$. In addition, the algorithm can conclude either $\lambda_{\min}(\nabla^2 f(\mathbf{x})) \geq -\varepsilon$ or find a unit vector $\mathbf{v}$ such that $\mathbf{v}^\top \nabla^2 f(\mathbf{x}) \mathbf{v} \leq -\varepsilon/2$. It can be implemented by runing $\log(1/\delta)$-copies Oja's algorithm (7) with a total $T = O\left(\frac{\log(d/\delta)^2}{\varepsilon^2}\right)$ iterations and $\eta = \Theta(\sqrt{T})$, and selecting one output from Oja's algorithm based on a boosting technique using an independent $T$ random $\nabla^2 f(\mathbf{x}; \xi)$ Hessian matrices.*

**Remark:** The above result was established in [4]. Please also refer to its proof of Lemma 3.3 in [4] for the boosting technique.

If the objective has a finite-sum structure $f(\mathbf{x}) = \frac{1}{m} \sum_{i=1}^m f_i(\mathbf{x})$, there also exist some stochastic algorithms that could have lower complexity than the Lanczos method or the method based on the Oja's algorithm.

**Algorithm 7** AdaNCD$^{\text{online}}$($\mathbf{x}, \alpha, \delta, \mathbf{g}(\mathbf{x})$):

1: Set $\varepsilon = \max(\epsilon_2, \|\mathbf{g}(\mathbf{x})\|^\alpha)/2$
2: Apply NCS($f, \mathbf{x}, \varepsilon, \delta$) to find a unit vector $\mathbf{v}$ that satisfies Lemma 7
3: **if** $\mathbf{v}^\top \nabla^2 f(\mathbf{x})\mathbf{v} \leq -\varepsilon/2$ and $\frac{\varepsilon^3}{24L_2^2} > \frac{\|\mathbf{g}(\mathbf{x})\|^2}{4L_1} - \frac{\epsilon'^2}{L_1}$ **then**
4:    Compute $\mathbf{x}^+ = \mathbf{x} - \frac{\varepsilon}{2L_2}z\mathbf{v}$
5: **else**
6:    Compute $\mathbf{x}^+ = \mathbf{x} - \frac{1}{L_1}\mathbf{g}(\mathbf{x})$
7: **end if**
8: **Return** $\mathbf{x}^+, \mathbf{v}$

---

**Lemma 6.** *There exists a randomized algorithm $\mathcal{A}$ such that with probability at least $1-\delta$, $\mathcal{A}$ produces a unit vector $\mathbf{v}$ satisfying (3) with a time complexity of $T_n(f, \varepsilon, \delta, d) = \widetilde{O}(d(m + m^{3/4}\sqrt{1/\varepsilon}))$.*
**Remark:** The randomized algorithms proposed in [5, 6] can serve this purpose.

*Proof.* We first introduce a proposition, which is the Theorem 2.5 in [7].

**Proposition 1.** *Let $M \in \mathbb{R}^{d \times d}$ be a symmetric matrix with eigenvalues $1 \geq \lambda_1 \ldots \geq \lambda_d \geq 0$. Then with probability at least $1 - p$, the Algorithm AppxPCA produces a unit vector $\mathbf{v}$ such that $\mathbf{v}^\top M \mathbf{v} \geq (1-\delta_+)(1-\epsilon)\lambda_{max}(M)$. The total running time is $\widetilde{O}\left(T_h^1 \max\{m, \frac{m^{3/4}}{\sqrt{\epsilon}}\} \log^2\left(\frac{1}{\epsilon^2 \delta_+}\right)\right)$.*

Define $M = I - \frac{H}{L_1}$, then $M$ satisfies the condition in the Proposition 1. Then we know that with probability at least $1 - p$, the Algorithm AppxPCA produces a vector $\mathbf{v}$ satisfying

$$\mathbf{v}^\top \left(I - \frac{H}{L_1}\right)\mathbf{v} \geq (1-\delta_+)(1-\epsilon)\left(1 - \frac{\lambda_{\min}(H)}{L_1}\right),$$

which implies that

$$L_1 - \mathbf{v}^\top H\mathbf{v} \geq (1 - \delta_+ - \epsilon + \delta_+\epsilon)(L_1 - \lambda_{\min}(H)) \geq (1 - \delta_+ - \epsilon)(L_1 - \lambda_{\min}(H)).$$

By simple algebra, we have

$$\lambda_{\min}(H) \geq \mathbf{v}^\top H\mathbf{v} - (\delta_+ + \epsilon)(L_1 - \lambda_{\min}(H)) \geq \mathbf{v}^\top H\mathbf{v} - 2L_1(\delta_+ + \epsilon).$$

By setting $\epsilon = \delta_+ = \frac{\varepsilon}{4L_1}$, we can finish the proof. $\qquad\square$

---

A standard NCD step is to update the solution by $\mathbf{x}^+ = \mathbf{x} - \eta\mathbf{v}$ with $\mathbf{v}$ being a negative curvature direction, where $\eta$ is a proper step size (e.g., see [8]). Almost all previous algorithms using NCD ask for a unit vector $\mathbf{v}$ to satisfy (3) with a noise level $\varepsilon = \Theta(\epsilon_2)$ whenever it is invoked.

## 2 Useful Lemmas for Adaptive Negative Curvature Step for Stochastic Objective

**Lemma 7.** *When $\lambda_{\min}(\nabla^2 f(\mathbf{x})) \leq -\varepsilon$, the Algorithm 7 provides a guarantee that*

$$f(\mathbf{x}) - \mathbb{E}[f(\mathbf{x}^+)] \geq \max\left\{\frac{\varepsilon^3}{24L_2^2}, \frac{\|\mathbf{g}(\mathbf{x})\|^2}{4L_1} - \frac{\epsilon'^2}{L_1}\right\}.$$

*Proof.* Since $f(\mathbf{x})$ has a $L_2$-Lipschitz continuous Hessian, we have

$$\left|f(\mathbf{x}_1) - f(\mathbf{x}) + \eta\mathbf{v}^\top \nabla f(\mathbf{x}) - \frac{1}{2}\eta^2\mathbf{v}^\top \nabla^2 f(\mathbf{x})\mathbf{v}\right| \leq \frac{L_2}{6}\|\eta\mathbf{v}\|^3.$$

When $\eta = \frac{\varepsilon}{2L_2}z$, define $\mathbf{x}_1 = \mathbf{x} - \eta\mathbf{v}$, where $\Pr(z = 1) = \Pr(z = -1) = \frac{1}{2}$, $\mathbf{v}$ is a unit vector and $\mathbf{v}^\top \nabla f(\mathbf{x})\mathbf{v} \leq -\frac{\varepsilon}{2}$. Note that $\mathbb{E}(\eta) = 0$ and $\mathbb{E}(\eta^2) = \frac{\varepsilon^2}{4L_2^2}$, then we have

$$f(\mathbf{x}) - \mathbb{E}(f(\mathbf{x}_1)) \geq \mathbb{E}\left(\eta\mathbf{v}^\top \nabla f(\mathbf{x}) - \frac{1}{2}\eta^2\mathbf{v}^\top \nabla^2 f(\mathbf{x})\mathbf{v} - \frac{L_2}{6}\|\eta\mathbf{v}\|^3\right) \geq \frac{\varepsilon^2}{8L_2^2} \cdot \frac{\varepsilon}{2} - \frac{L_2}{6} \cdot \frac{\varepsilon^3}{8L_2^3} = \frac{\varepsilon^3}{24L_2^2}.$$

Define $\mathbf{x}_2 = \mathbf{x} - \frac{1}{L_1}\mathbf{g}(\mathbf{x})$, where $\|\mathbf{g}(\mathbf{x}) - \nabla f(\mathbf{x})\| \le \epsilon'$, and then we have

$$
\begin{aligned}
f(\mathbf{x}_2) - f(\mathbf{x}) &\le (\mathbf{x}_2 - \mathbf{x})^\top \nabla f(\mathbf{x}) + \frac{L_1}{2}\|\mathbf{x}_2 - \mathbf{x}\|^2 \\
&= -\frac{1}{L_1}\mathbf{g}(\mathbf{x})^\top \nabla f(\mathbf{x}) + \frac{\|\mathbf{g}(\mathbf{x})\|^2}{2L_1} \\
&= -\frac{1}{L_1}\mathbf{g}(\mathbf{x})^\top \mathbf{g}(\mathbf{x}) + \frac{1}{L_1}\mathbf{g}(\mathbf{x})^\top(\mathbf{g}(\mathbf{x}) - \nabla f(\mathbf{x})) + \frac{\|\mathbf{g}(\mathbf{x})\|^2}{2L_1} \\
&\le -\frac{1}{2L_1}\|\mathbf{g}(\mathbf{x})\|^2 + \frac{1}{4L_1}\|\mathbf{g}(\mathbf{x})\|^2 + \frac{1}{L_1}\|\mathbf{g}(\mathbf{x}) - \nabla f(\mathbf{x})\|^2 \\
&= -\frac{1}{4L_1}\|\mathbf{g}(\mathbf{x})\|^2 + \frac{\epsilon'^2}{L_1}.
\end{aligned}
$$

Combining two cases ($\mathbf{x}_1$ and $\mathbf{x}_2$, which correspond to line 4 and line 6 of Algorithm 2 respectively), here completes the proof. $\square$

**Lemma 8.** *For any $\epsilon > 0, \delta' \in (0,1)$, $\mathbf{x} \in \mathbb{R}^d$, when elements of $\mathcal{S}$ are uniformly selected from $\{1,\ldots, n\}$ with $|\mathcal{S}| \ge \frac{16L_1^2}{\epsilon^2}\log(\frac{2d}{\delta'})$, we have*

$$\Pr(\|H_{\mathcal{S}}(\mathbf{x}) - \nabla^2 f(\mathbf{x})\|_2 \le \epsilon) \ge 1 - \delta'.$$

The above lemma can be proved by using matrix concentration inequalities. Please see [9][Lemma 4] for a proof.

**Lemma 9.** *Assume that $\mathbb{E}[\exp(\|\nabla f(\mathbf{x};\xi) - \nabla f(\mathbf{x})\|^2 / G^2)] \le \exp(1)$ holds for any $\mathbf{x} \in \mathbb{R}^d$. For any $\epsilon > 0$, $\delta' \in (0,1)$, $\mathbf{x} \in \mathbb{R}^d$, when $|\mathcal{S}_1| \ge \frac{4G^2(1+3\log(1/\delta))}{\epsilon^2}$, we have*

$$\Pr(\|\mathbf{g}(\mathbf{x}) - \nabla f(\mathbf{x})\| \le \epsilon) \ge 1 - \delta'.$$

*where $\mathcal{S}_1$ a set of random samples $\xi$, due to the exponential tail behavior of stochastic gradients.*

**Remark:** Lemma 9 can be proved by using large deviation theorem of vector-valued martingales (e.g., see [10][Lemma 4]).

# 3 Proof of Lemma 1

The Proof of Lemma 1 can be derived by combining the result of Lemma 4, 5 and 6.

# 4 Proof of Lemma 2

*Proof.* Denote $\eta = \frac{2|\mathbf{v}^\top \nabla^2 f(\mathbf{x})\mathbf{v}|}{L_2}\mathrm{sign}(\mathbf{v}^\top \nabla f(\mathbf{x}))$ with $\|\mathbf{v}\| = 1$. Let $\mathbf{x}_1^+ = \mathbf{x} - \eta\mathbf{v}$ denote the updated solution if following $\mathbf{v}$ and $\mathbf{x}_2^+ = \mathbf{x} - \nabla f(\mathbf{x})/L_1$ denote the updated solution if following $\nabla f(\mathbf{x})$. Since $f(\mathbf{x})$ has a $L_2$-Lipschitz continuous Hessian, we have

$$|f(\mathbf{x}_1^+) - f(\mathbf{x}) + \eta\mathbf{v}^\top\nabla f(\mathbf{x}) - \frac{1}{2}\eta^2\mathbf{v}^\top\nabla^2 f(\mathbf{x})\mathbf{v}| \le \frac{L_2}{6}\|\eta\mathbf{v}\|^3.$$

By noting that $\eta\mathbf{v}^\top\nabla f(\mathbf{x}) \ge 0$ and when $\mathbf{v}^\top\nabla^2 f(\mathbf{x})\mathbf{v} \le 0$, we have

$$f(\mathbf{x}) - f(\mathbf{x}_1^+) \ge -\frac{1}{2}\eta^2\mathbf{v}^\top\nabla^2 f(\mathbf{x})\mathbf{v} - \frac{L_2}{6}\|\eta\mathbf{v}\|^3 = \frac{2(-\mathbf{v}^\top\nabla^2 f(\mathbf{v})\mathbf{v})^3}{3L_2^2} \triangleq \Delta_1.$$

By the smoothness of $f(\mathbf{x})$, we have

$$
\begin{aligned}
f(\mathbf{x}_2^+) &\le f(\mathbf{x}) + \nabla f(\mathbf{x})^\top(\mathbf{x}_2^+ - \mathbf{x}) + \frac{L_1}{2}\|\mathbf{x}_2^+ - \mathbf{x}\|^2 \\
&= f(\mathbf{x}) - \frac{1}{L_1}\|\nabla f(\mathbf{x})\|_2^2 + \frac{L_1\eta^2}{2}\|\nabla f(\mathbf{x})\|^2 \\
&\le f(\mathbf{x}) - \frac{1}{2L_1}\|\nabla f(\mathbf{x})\|^2
\end{aligned}
$$

As a result, $f(\mathbf{x}) - f(\mathbf{x}_2^+) \geq \frac{\|\nabla f(\mathbf{x})\|^2}{2L_1} \triangleq \Delta_2$.

According to the update rule in AdaNCD$^{\text{det}}$ (Algorithm 1), if $\Delta_1 > \Delta_2$, we have $\mathbf{x}^+ = \mathbf{x}_1^+$ and $f(\mathbf{x}) - f(\mathbf{x}^+) \geq \Delta_1 = \max(\Delta_1, \Delta_2)$. If $\Delta_2 \geq \Delta_1$, then $\mathbf{x}^+ = \mathbf{x}_2^+$ and $f(\mathbf{x}) - f(\mathbf{x}^+) \geq \Delta_2 = \max(\Delta_1, \Delta_2)$. In both cases, we have $f(\mathbf{x}) - f(\mathbf{x}^+) \geq \max(\Delta_1, \Delta_2)$. $\qquad\square$

## 5 Proof of Lemma 3

*Proof.* Define $\eta = \frac{2|\mathbf{v}^\top H_{\mathcal{S}}(\mathbf{x})\mathbf{v}|}{L_2} z$, $\mathbf{x}_1 = \mathbf{x} - \eta\mathbf{v}$, where $\Pr(z = 1) = \Pr(z = -1) = \frac{1}{2}$, $\mathbf{v}$ is a unit vector and $\mathbf{v}^\top H_{\mathcal{S}}(\mathbf{x})\mathbf{v} \leq 0$. Note that $\mathbb{E}(\eta) = 0$ and $\mathbb{E}(\eta^2) = \frac{4|\mathbf{v}^\top H_{\mathcal{S}}(\mathbf{x})\mathbf{v}|^2}{L_2^2}$, then by the $L_2$-Lipschitz continuous Hessian, we have

$$f(\mathbf{x}) - \mathbb{E}(f(\mathbf{x}_1))$$

$$\geq \mathbb{E}\left(\eta\mathbf{v}^\top \nabla f(\mathbf{x}) - \frac{1}{2}\eta^2\mathbf{v}^\top \nabla^2 f(\mathbf{x})\mathbf{v} - \frac{L_2}{6}\|\eta\mathbf{v}\|^3\right)$$

$$= -\frac{2|\mathbf{v}^\top H_{\mathcal{S}}(\mathbf{x})\mathbf{v}|^2}{L_2^2}\left(\mathbf{v}^\top(\nabla^2 f(\mathbf{x}) - H_{\mathcal{S}}(\mathbf{x}))\mathbf{v}\right) - \frac{2|\mathbf{v}^\top H_{\mathcal{S}}(\mathbf{x})\mathbf{v}|^2}{L_2^2}\mathbf{v}^\top H_{\mathcal{S}}(\mathbf{x})\mathbf{v} - \frac{L_2}{6} \cdot \frac{8|\mathbf{v}^\top H_{\mathcal{S}}(\mathbf{x})\mathbf{v}|^3}{L_2^3}$$

$$\geq \frac{2|\mathbf{v}^\top H_{\mathcal{S}}(\mathbf{x})\mathbf{v}|^3}{3L_2^2} - \frac{\epsilon_2|\mathbf{v}^\top H_{\mathcal{S}}(\mathbf{x})\mathbf{v}|^2}{6L_2^2}$$

where the last inequality holds because of the inequality (7).

Define $\mathbf{x}_2 = \mathbf{x} - \frac{1}{L_1}\mathbf{g}(\mathbf{x})$, where $\|\mathbf{g}(\mathbf{x}) - \nabla f(\mathbf{x})\| \leq \epsilon'$. By the same argument as in the proof of Lemma 6, we have

$$f(\mathbf{x}) - f(\mathbf{x}_2) \geq \frac{\|\mathbf{g}(\mathbf{x})\|^2}{4L_1} - \frac{\epsilon'^2}{L_1}.$$

Combining two cases ($\mathbf{x}_1$ and $\mathbf{x}_2$, which correspond to line 3 and line 5 of Algorithm 3 respectively), we have

$$f(\mathbf{x}) - \mathbb{E}[f(\mathbf{x}^+)] \geq \max\left\{\frac{-2(\mathbf{v}^\top H_{\mathcal{S}}(\mathbf{x})\mathbf{v})^3}{3L_2^2} - \frac{\epsilon_2|\mathbf{v}^\top H_{\mathcal{S}}(\mathbf{x})\mathbf{v}|^2}{6L_2^2}, \frac{\|\mathbf{g}(\mathbf{x})\|^2}{4L_1} - \frac{\epsilon'^2}{L_1}\right\}$$

The result when $\mathbf{v}^\top H_{\mathcal{S}}(\mathbf{x})\mathbf{v} \leq -\epsilon_2/2$ directly follows from the above inequality. $\qquad\square$

## 6 Proof of Theorem 1

*Proof.* Define $\epsilon_2 = \epsilon_1^\alpha$. Let $j_*$ denote the $j$ such that the algorithm terminates. Then for all $j < j_*$, we have $\|\nabla f(\mathbf{x}_j)\| > \epsilon_1$, or $\mathbf{v}_j^\top \nabla^2 f(\mathbf{x}_j)\mathbf{v}_j \leq -\epsilon_2/2$. According to Lemma 4, we have

$$f(\mathbf{x}_j) - f(\mathbf{x}_{j+1}) \geq \max\left(\frac{2|\mathbf{v}_j^\top \nabla^2 f(\mathbf{x}_j)\mathbf{v}_j|^3}{3L_2^2}, \frac{\|\nabla f(\mathbf{x}_j)\|^2}{2L_1}\right)$$

Let us consider three cases. Case 1: $\|\nabla f(\mathbf{x}_j)\| > \epsilon_1$ and $\mathbf{v}_j^\top \nabla^2 f(\mathbf{x}_j)\mathbf{v}_j \leq -\epsilon_2/2$, then we have

$$\max\left(\frac{\epsilon_2^3}{12L_2^2}, \frac{\epsilon_1^2}{2L_1}\right) \leq f(\mathbf{x}_j) - f(\mathbf{x}_{j+1})$$

Case 2: $\|\nabla f(\mathbf{x}_j)\| \leq \epsilon_1$ and $\mathbf{v}_j^\top \nabla^2 f(\mathbf{x}_j)\mathbf{v}_j \leq -\epsilon_2/2$, we have

$$\frac{\epsilon_2^3}{12L_2^2} \leq f(\mathbf{x}_j) - f(\mathbf{x}_{j+1})$$

Case 3: $\|\nabla f(\mathbf{x}_j)\| > \epsilon_1$ and $\mathbf{v}_j^\top \nabla^2 f(\mathbf{x}_j)\mathbf{v}_j > -\epsilon_2/2$, we have

$$\frac{\epsilon_1^2}{2L_1} \leq f(\mathbf{x}_j) - f(\mathbf{x}_{j+1})$$

In any case, we have

$$\min\left(\frac{\epsilon_1^2}{2L_1}, \frac{\epsilon_2^3}{12L_2^2}\right) \leq f(\mathbf{x}_j) - f(\mathbf{x}_{j+1})$$

Then with at most $j_* = 1 + \max\left(\frac{12L_2^2}{\epsilon_2^3}, \frac{2L_1}{\epsilon_1^2}\right)\Delta$, the algorithm terminates. Note that $\epsilon_2 = \epsilon_1^\alpha$, we know that $j_* = 1 + \max\left(\frac{12L_2^2}{\epsilon_1^{3\alpha}}, \frac{2L_1}{\epsilon_1^2}\right)\Delta$.

Upon termination, we have with probability at least $1 - j_*\delta'$, i.e. with probability at least $1 - \delta$,

$$\lambda_{\min}(\nabla^2 f(\mathbf{x}_{j_*})) \geq -\epsilon_2/2 - \max(\epsilon_2, \|\nabla f(\mathbf{x}_{j_*})\|^\alpha)/2$$
$$= -\epsilon_1^\alpha/2 - \max(\epsilon_1^\alpha, \|\nabla f(\mathbf{x}_{j_*})\|^\alpha)/2.$$

Since $\|\nabla f(\mathbf{x}_{j_*})\| \leq \epsilon_1$, we have

$$\max(\epsilon_1^\alpha, \|\nabla f(\mathbf{x}_{j_*})\|^\alpha) = \epsilon_1^\alpha,$$

and hence $\lambda_{\min}(\nabla^2 f(\mathbf{x}_{j_*})) \geq -\epsilon_1^\alpha$.

The running time spent on the $j$-th iteration follows from Lemma 1. □

# 7   Proof of Theorem 2

For the $j$-th AdaNCD$^{\text{mb}}$ step, define the event $\mathcal{A} = \{\|H(\mathbf{x}_j) - \nabla^2 f(\mathbf{x}_j)\|_2 \leq \epsilon_2/6\} \cap \{\|\mathbf{g}(\mathbf{x}_j) - \nabla f(\mathbf{x}_j)\| \leq \epsilon_1/2\sqrt{2}\}$ and let $\Pr(\mathcal{A}) = 1 - \delta'$. Since the Algorithm S-AdaNCG calls AdaNCD$^{\text{mb}}$ as a subroutine, then by Lemma 8, when $\mathbf{v}_j^\top H_{\mathcal{S}_2}(\mathbf{x}_j)\mathbf{v}_j \leq -\epsilon_2/2$ with probability at least $1 - \delta'$,

$$f(\mathbf{x}_j) - \mathbb{E}[f(\mathbf{x}_{j+1})] \geq \max\left(\frac{1}{4L_1}\|\mathbf{g}(\mathbf{x}_j)\|^2 - \frac{\epsilon_1^2}{8L_1}, \frac{-2(\mathbf{v}^\top H_{\mathcal{S}}(\mathbf{x})\mathbf{v})^3}{3L_2^2} - \frac{\epsilon_2|\mathbf{v}^\top H_{\mathcal{S}}(\mathbf{x})\mathbf{v}|^2}{6L_2^2}\right).$$

If $\mathbf{v}_j^\top H_{\mathcal{S}_2}(\mathbf{x}_j)\mathbf{v}_j \leq -\epsilon_2/2$, we have

$$f(\mathbf{x}_j) - \mathbb{E}[f(\mathbf{x}_{j+1})] \geq \frac{|\mathbf{v}_j^\top H_{\mathcal{S}}(\mathbf{x}_j)\mathbf{v}_j|^2(-4\mathbf{v}_j^\top H_{\mathcal{S}}(\mathbf{x}_j)\mathbf{v}_j - \epsilon_2)}{6L_2^2} \geq \frac{|\mathbf{v}_j^\top H_{\mathcal{S}}(\mathbf{x}_j)\mathbf{v}_j|^2\epsilon_2}{6L_2^2} \geq \frac{\epsilon_2^3}{24L_2^2}$$

If $\|\mathbf{g}(\mathbf{x}_j)\| > \epsilon_1$, we have

$$f(\mathbf{x}_j) - \mathbb{E}[f(\mathbf{x}_{j+1})] \geq \frac{\epsilon_1^2}{8L_1}$$

Following the boosting argument in [11][Theorem 14], with high probability $1 - \zeta$ the algorithm terminates after $O(\log(1/\zeta)\max(1/\epsilon_1^2, 1/\epsilon_2^3))$ steps with high probability. Upon termination at iteration $j_*$ we have $\mathbf{v}_{j_*}^\top H(\mathbf{x}_{j_*})\mathbf{v}_{j_*} \geq -\epsilon_2/2$ and $\|\mathbf{g}(\mathbf{x}_{j_*})\| \leq \epsilon_1$. Next, we show that upon termination, we achieve an $(2\epsilon_1, 2\epsilon_2)$-second order stationary point with high probability. In particular, with probability $1 - \delta'$ we have

$$\|\nabla f(\mathbf{x}_{j_*})\| \leq \|\nabla f(\mathbf{x}_{j_*}) - \mathbf{g}(\mathbf{x}_{j_*})\| + \|\mathbf{g}(\mathbf{x}_{j_*})\| \leq \epsilon_1/2\sqrt{2} + \epsilon_1 \leq 2\epsilon_1.$$

and with probability $1 - \delta'$

$$\lambda_{\min}(H(\mathbf{x}_{j_*})) \geq \mathbf{v}_{j_*}^\top H(\mathbf{x}_{j_*})\mathbf{v}_{j_*} - \max(\epsilon_2, \|g(\mathbf{x}_{j_*})\|^\alpha)/2 \geq -\epsilon_2$$

In addition, with probability $1 - \delta'$, we have

$$\lambda_{\min}(\nabla^2 f(\mathbf{x}_{j_*})) \geq \lambda_{\min}(H(\mathbf{x}_{j_*})) - \epsilon_2/12 \geq -2\epsilon_2$$

As a result, by using union bound, we have with probability $1 - 3j_*\delta' = 1 - 3\delta$, we have

$$\|\nabla f(\mathbf{x}_{j_*})\| \leq 2\epsilon_1, \quad \lambda_{\min}(\nabla^2 f(\mathbf{x}_{j_*})) \geq -2\epsilon_2$$

# 8 Proof of Theorem 3

Before diving into the proofs, we first present the procedure Almost-Convex-AGD (Algorithm 8) and introduce some propositions which are useful for our further analysis.

---

**Algorithm 8** Almost-Cvx-AGD$(f, \mathbf{z}_1, \epsilon, \gamma, L_1)$

---

1: **for** $j = 1, 2, \ldots$ **do**
2:     **if** $\|\nabla f(\mathbf{z}_j)\| \leq \epsilon$ **then**
3:        **Return** $\mathbf{z}_j$
4:     **end if**
5:     Define $g_j(\mathbf{z}) = f(\mathbf{z}) + \gamma\|\mathbf{z} - \mathbf{z}_j\|^2$
6:     set $\epsilon' = \epsilon\sqrt{\gamma/50(L_1 + 2\gamma)}$
7:     $\mathbf{z}_{j+1} = \text{AGD}(g_j, \mathbf{z}_j, \epsilon', L_1, \gamma)$
8: **end for**

---

---

**Algorithm 9** AGD$(f, \mathbf{y}_1, \epsilon, L_1, \sigma_1)$

---

1: Set $\kappa = L_1/\sigma_1$, $\mathbf{z}_1 = \mathbf{y}_1$
2: **for** $j = 1, 2, \ldots$ **do**
3:     **if** $\|\nabla f(\mathbf{y}_j)\| \leq \epsilon$ **then**
4:        **Return** $\mathbf{y}_j$
5:     **end if**
6:     $\mathbf{y}_{j+1} = \mathbf{z}_j - \frac{1}{L_1}\nabla f(\mathbf{z}_j)$
7:     $\mathbf{z}_{j+1} = (1 + \frac{\sqrt{\kappa}-1}{\sqrt{\kappa}+1})\mathbf{y}_{j+1} - \frac{\sqrt{\kappa}-1}{\sqrt{\kappa}+1}\mathbf{y}_j$
8: **end for**

---

**Proposition 2** (Lemma 3.1 of [8]). *Let* $f : \mathbb{R}^d \to \mathbb{R}$ *be* $\gamma$-*almost convex and* $L_1$-*smooth, where* $0 < \gamma \leq L_1$. *Then Almost-Convex-AGD$(f, \mathbf{z}_1, \epsilon, \gamma, L_1)$ returns a vector* $\mathbf{z}$ *such that* $\|\nabla f(\mathbf{z})\| \leq \epsilon$ *and*

$$f(\mathbf{z}_1) - f(\mathbf{z}) \geq \min\left\{\gamma\|\mathbf{z} - \mathbf{z}_1\|^2, \frac{\epsilon}{\sqrt{10}\|\mathbf{z} - \mathbf{z}_1\|}\right\} \tag{8}$$

*in time*

$$O\left(T_g\left(\sqrt{\frac{L_1}{\gamma}} + \frac{\sqrt{\gamma L_1}}{\epsilon^2}(f(\mathbf{z}_1) - f(\mathbf{z}))\right)\log\left(2 + \frac{L_1^3\Delta}{\gamma^2\epsilon^2}\right)\right) \tag{9}$$

**Proposition 3** (Lemma 4.1 of [8]). *Let* $f$ *be* $L_1$-*smooth and have* $L_2$-*Lipschitz continuous Hessian. Let* $\mathbf{x}_0 \in \mathbb{R}^d$ *be such that* $\nabla^2 f(\mathbf{x}_0) \succeq -\alpha I$ *for some* $\alpha \geq 0$, *then* $f(\mathbf{x}) + L_1\left[\|\mathbf{x}\| - \frac{\alpha}{L_2}\right]_+$ *is* $3\alpha$-*almost convex and* $5L_1$-*smooth.*

The next result is a corollary of Theorem 1, showing that by running $\widehat{\mathbf{x}}_k = \text{AdaNCG}(\mathbf{x}_k, \epsilon_2^{3/2}, \epsilon_2, \delta')$ we obtain a solution $\widehat{\mathbf{x}}_k$ around which $f(\mathbf{x})$ is locally almost convex, i.e., $\nabla^2 f(\widehat{\mathbf{x}}_k) \succeq -\epsilon_2 I$.

**Corollary 1.** *The sub-routine* $\widehat{\mathbf{x}}_k = AdaNCG(\mathbf{x}_k, \epsilon_2^{3/2}, \epsilon_2, \delta')$ *guarantees that*

$$\lambda_{\min}(\nabla^2 f(\widehat{\mathbf{x}}_k)) \geq -\epsilon_2$$

*with at most* $j_k$ *iterations within AdaNCG, where*

$$j_k \leq 1 + \frac{\max(12L_2^2, 2L_1)}{\epsilon_2^3}(f(\mathbf{x}_k) - f(\widehat{\mathbf{x}}_k)) \leq 1 + \frac{\max(12L_2^2, 2L_1)}{\epsilon_2^3}\Delta, \tag{10}$$

*Furthermore, each iteration* $j$ *within AdaNCG requires time at most*

$$O\left(T_h\frac{\sqrt{L_1}}{\max(\epsilon_2, \|\nabla f(\mathbf{x}_j)\|)^{1/2}}\log\left(\frac{d}{\delta'}\right)\right)$$

*Proof of Theorem 3.* We try to bound the number of iterations in the Algorithm AdaNCG$^+$, which is actually the upper bound of the number of calls of both AdaNCG and Almost-Convex-AGD.

Define $\rho_\alpha(\mathbf{x}) := L_1 \left[ \|\mathbf{x}\| - \frac{\alpha}{L_2} \right]_+$. At iteration $k$ when $\|\nabla f(\widehat{\mathbf{x}}_k)\| \leq \epsilon_1$ is not met, which means $\|\nabla f(\widehat{\mathbf{x}}_k)\| > \epsilon_1$, we have

$$\epsilon_1 < \|\nabla f(\widehat{\mathbf{x}}_k)\| \leq [\|\nabla f_{k-1}(\widehat{\mathbf{x}}_k)\| + \|\nabla \rho_{\epsilon_2}(\widehat{\mathbf{x}}_k - \widehat{\mathbf{x}}_{k-1})\|] \leq \frac{\epsilon_1}{2} + 2L_1 \left[ \|\widehat{\mathbf{x}}_k - \widehat{\mathbf{x}}_{k-1}\| - \frac{\epsilon_2}{L_2} \right]_+,$$

where the second inequality holds due to the triangle inequality, and the third inequality holds because of the guarantee provided by Almost-Convex-AGD at the previous stage. Therefore, we have

$$\frac{\epsilon_1}{4L_1} \leq \left[ \|\widehat{\mathbf{x}}_k - \widehat{\mathbf{x}}_{k-1}\| - \frac{\epsilon_2}{L_2} \right]_+ = \|\widehat{\mathbf{x}}_k - \widehat{\mathbf{x}}_{k-1}\| - \frac{\epsilon_2}{L_2} \tag{11}$$

According to the inequality (11), we know that at iteration $1 < k \leq K$, exactly one of the following three cases is true:

    (I) $\|\nabla f(\widehat{\mathbf{x}}_k)\| \leq \epsilon_1$ and the Algorithm AdaNCG$^+$ terminates

    (II) $\|\nabla f(\widehat{\mathbf{x}}_k)\| > \epsilon_1$ (which implies that $\|\widehat{\mathbf{x}}_k - \widehat{\mathbf{x}}_{k-1}\| \geq \frac{\epsilon_2}{L_2}$ according to (11)), and $\widehat{\mathbf{x}}_k \neq \mathbf{x}_k$

    (III) $\|\nabla f(\widehat{\mathbf{x}}_k)\| > \epsilon_1$ and $\widehat{\mathbf{x}}_k = \mathbf{x}_k$

If (II) holds, note that the subroutine AdaNCG needs at least 2 iterations, so according to Theorem 1, we have

$$\max \left( \frac{12L_2^2}{\epsilon_2^3}, \frac{2L_1}{\epsilon_2^3} \right) (f(\mathbf{x}_k) - f(\widehat{\mathbf{x}}_k)) \geq 1.$$

Combining it with the progressive bound (8) in Proposition 2, we have

$$f(\widehat{\mathbf{x}}_{k-1}) - f(\widehat{\mathbf{x}}_k) \geq f(\mathbf{x}_k) - f(\widehat{\mathbf{x}}_k) \geq \min \left( \frac{\epsilon_2^3}{12L_2^2}, \frac{\epsilon_2^3}{2L_1} \right).$$

If (III) holds, then by Proposition 3 and the second-order guarantee provided by Theorem 1, we can know that, with probability at least $1 - \delta'$, $f_k$ is $3\epsilon_2$-almost convex and $5L_1$-smooth. Then applying Proposition 2 suffices to show that

$$f(\widehat{\mathbf{x}}_{k-1}) - f(\widehat{\mathbf{x}}_k) \geq \min \left\{ 3\epsilon_2 \|\widehat{\mathbf{x}}_{k-1} - \mathbf{x}_k\|^2, \frac{\epsilon_1}{2\sqrt{10}} \|\widehat{\mathbf{x}}_{k-1} - \mathbf{x}_k\| \right\} \geq \min \left\{ \frac{3\epsilon_2^3}{L_2^2}, \frac{\epsilon_1 \epsilon_2}{2\sqrt{10}L_2} \right\}.$$

Combing two cases (II) and (III) together, we get the conclusion that whether in case (II) or case (III), with probability at least $1 - \delta'$,

$$f(\widehat{\mathbf{x}}_{k-1}) - f(\widehat{\mathbf{x}}_k) \geq \min \left( \frac{\epsilon_2^3}{12L_2^2}, \frac{\epsilon_2^3}{2L_1}, \frac{\epsilon_1 \epsilon_2}{2\sqrt{10}L_2} \right).$$

In order to get a contradiction that after $K$ iterations the algorithm has not terminated yet, and by the definition of $\delta'$ and union bound, it follows that, with probability at least $1 - \delta$,

$$\Delta \geq f(\widehat{\mathbf{x}}_1) - f(\widehat{\mathbf{x}}_K) = \sum_{k=1}^{K-1} (f(\widehat{\mathbf{x}}_k) - f(\widehat{\mathbf{x}}_{k+1})) \geq (K-1) \min \left( \frac{\epsilon_2^3}{12L_2^2}, \frac{\epsilon_2^3}{2L_1}, \frac{\epsilon_1 \epsilon_2}{2\sqrt{10}L_2} \right).$$

Plugging in $K = \lceil 1 + \Delta \left( \frac{\max(12L_2^2, 2L_1)}{\epsilon_2^3} + \frac{2\sqrt{10}L_2}{\epsilon_1 \epsilon_2} \right) \rceil$ suffices to get a contradiction. Therefore the algorithm terminates after at most $K$ outer iterations.

Denote $T_g$ and $T_h$ by the time for gradient evaluation and Hessian-vector product evaluation. Define $\tau = 1 + 1/\epsilon + 1/\delta + d + L_1 + L_2 + \Delta$. We try to bound the number of AdaNCD$^{\text{det}}$ steps. Denote $j_k$ by the total number of times the Algorithm AdaNCG is executed during the

iteration $k$ of the method AdaNCG$^+$, and define $k^*$ as the total number of outer iterations of the Algorithm AdaNCG$^+$. By telescoping bound (10) and the progressive bound (8) of Proposition 2 in Almost-Convex-AGD, which guarantees the Almost-Convex-AGD decreases the function values, we have

$$\sum_{k=1}^{k^*}(j_k - 1) \leq \sum_{k=1}^{k^*} \max\left(\frac{12L_2^2}{\epsilon_2^3}, \frac{2L_1}{\epsilon_2^3}\right)(f(\mathbf{x}_k) - f(\widehat{\mathbf{x}}_k))$$

$$\leq \sum_{k=1}^{k^*} \max\left(\frac{12L_2^2}{\epsilon_2^3}, \frac{2L_1}{\epsilon_2^3}\right)(f(\widehat{\mathbf{x}}_{k-1}) - f(\widehat{\mathbf{x}}_k))$$

$$\leq \max\left(\frac{12L_2^2}{\epsilon_2^3}, \frac{2L_1}{\epsilon_2^3}\right)\Delta.$$

According to the previous result, with probability at least $1 - \delta$, we can have a upper bound of $k^*$, which is

$$k^* \leq 2 + \Delta\left(\frac{12L_2^2}{\epsilon_2^3} + \frac{2L_1}{\epsilon_2^3} + \frac{2\sqrt{10}L_2}{\epsilon_1\epsilon_2}\right). \tag{12}$$

Hence, we have with probability at least $1 - \delta$,

$$\sum_{k=1}^{k^*} j_k = k^* + \sum_{k=1}^{k^*}(j_k - 1)$$
$$\leq 2 + \Delta\left(\frac{24L_2^2}{\epsilon_2^3} + \frac{4L_1}{\epsilon_2^3} + \frac{2\sqrt{10}L_2}{\epsilon_1\epsilon_2}\right). \tag{13}$$

According to Corollary 1, we have the cost of each iteration $t$ within AdaNCG is

$$O\left(T_h \frac{\sqrt{L_1}}{\max(\epsilon_2, \|\nabla f(\mathbf{x}_t)\|)^{1/2}} \log\left(\frac{d}{\delta'}\right)\right).$$

Note that the failure probability satisfies

$$\frac{1}{\delta'} \leq \frac{2 + \Delta\left(\frac{12L_2^2}{\epsilon_2^3} + \frac{2L_1}{\epsilon_2^3} + \frac{2\sqrt{10}L_2}{\epsilon_1\epsilon_2}\right)}{\delta},$$

so $\log\frac{d}{\delta'} = O(\log\tau)$. Then we employ (13) to bound the worst-case total costs of AdaNCG, which is

$$O\left(T_h \frac{\sqrt{L_1}}{\sqrt{\epsilon_2}}\left[2 + \Delta\left(\frac{24L_2^2}{\epsilon_2^3} + \frac{4L_1}{\epsilon_2^3} + \frac{2\sqrt{10}L_2}{\epsilon_1\epsilon_2}\right)\right]\log\tau\right) \tag{14}$$

Now we analyze the total cost of calling Almost-Convex-AGD. Employing the bound (9) in Proposition 9, the cost of calling Almost-Convex-AGD in iteration $k$ with almost convexity parameter $3\epsilon_2$ is bounded by

$$O\left(T_g\left(\sqrt{\frac{L_1}{3\epsilon_2}} + \frac{4\sqrt{3\epsilon_2 L_1}}{\epsilon_1^2}[f_k(\mathbf{x}_k) - f_k(\mathbf{x}_{k+1})]\right)\log\tau\right).$$

Note that $f_k(\mathbf{x}_k) - f_k(\mathbf{x}_{k+1}) \leq f(\mathbf{x}_k) - f(\mathbf{x}_{k+1})$, so we have

$$\sum_{k=1}^{k^*}[f_k(\mathbf{x}_k) - f_k(\mathbf{x}_{k+1})] \leq \sum_{k=1}^{k^*}[f(\mathbf{x}_k) - f(\mathbf{x}_{k+1})] \leq \Delta.$$

According to (12), we can get that the total time complexity of Almost-Convex-AGD is

$$O\left(T_g(\xi_1 + \xi_2)\log\tau\right), \tag{15}$$

where $\xi_1 = \sqrt{\frac{L_1}{3\epsilon_2}}\left(2 + \Delta\left(\frac{24L_2^2}{\epsilon_2^3} + \frac{4L_1}{\epsilon_2^3} + \frac{2\sqrt{10}L_2}{\epsilon_1\epsilon_2}\right)\right)$, $\xi_2 = \frac{4\sqrt{3\epsilon_2 L_1}}{\epsilon_1^2}\Delta$. According to (14) and (15), and note that $T_g = T_h = O(d)$, we get that the worst case complexity bound is

$$\widetilde{O}\left(\left(\frac{1}{\epsilon_1\epsilon_2^{3/2}} + \frac{1}{\epsilon_2^{7/2}}\right)T_h + \frac{\epsilon_2^{1/2}}{\epsilon_1^2}T_g\right),$$

where $\widetilde{O}(\cdot)$ hides a $\log\tau$ factor. Since $T_g = T_h = O(d)$, the proof is complete. $\qquad\square$

**Algorithm 10** SCSG-Epoch: $(\mathbf{x}, \mathcal{S}, b)$

1: **Input**: $\mathbf{x}, \epsilon\, m_1, b$
2: Set $\eta = c'(m_1/b)^{-2/3}$ with $c' \leq 1/6$
3: Compute $\nabla F_{\mathcal{S}}(\mathbf{x}_{j-1})$
4: Let $\mathbf{x}_0 = \mathbf{x}$ and generate $N \sim \text{Geom}(m_1/(m_1 + b))$
5: **for** $k = 1, 2, \ldots, N$ **do**
6:   Sample samples $\mathcal{S}_k$ of size $b$
7:   Compute $\mathbf{v}_k = \nabla f_{\mathcal{S}_k}(\mathbf{x}_{k-1}) - \nabla f_{\mathcal{S}_k}(\mathbf{x}_0) + \nabla f_{\mathcal{S}}(\mathbf{x}_0)$
8:     $\mathbf{x}_k = \mathbf{x}_{k-1} - \eta \mathbf{v}_k$
9: **end for**
10: **Return** $\mathbf{x}_N$

# 9 Proof of Theorem 4

The proof of this Theorem follows that of Theorem 3 in [12]. Similarly, we prove the following two lemmas.

**Lemma 10.** *Suppose* $|\mathcal{S}| \geq O(1/\epsilon^2)$ *and* $|\mathcal{S}_2| \geq \widetilde{O}(1/\gamma^2)$. *For any point* $\mathbf{y}_j$ *with* $\|\nabla f(\mathbf{y}_j)\| \geq \epsilon$, *then we can have*

$$\mathbb{E}[f(\mathbf{x}_{j+1}) - f(\mathbf{x}_j)] \leq -\Omega(\epsilon^{4/3}).$$

*Proof.* Due to the update in AdaNCD$^{\text{mb}}$, it is clear that $\mathbb{E}[f(\mathbf{x}_{j+1}) - f(\mathbf{y}_j)] \leq 0$. Following the analysis of Lemma 7 in [12], we have $\mathbb{E}[f(\mathbf{y}_j) - f(\mathbf{x}_j)] \leq -\Omega(\epsilon^{4/3})$. $\square$

**Lemma 11.** *Suppose* $|\mathcal{S}| \geq O\left(\frac{1}{\epsilon_2^{9/2}b^{1/2}}\right), |\mathcal{S}_2| \geq \widetilde{O}(1/\epsilon_2^2)$. *For any point* $\mathbf{y}_j$ *with* $\|\nabla f(\mathbf{y}_j)\| \leq \epsilon$ *and* $\mathbf{v}_j^\top H_{\mathcal{S}_2}(\mathbf{y}_j)\mathbf{v}_j \leq -\epsilon_2/2$, *we can have*

$$\mathbb{E}[f(\mathbf{x}_{j+1}) - f(\mathbf{x}_j)] \leq -\widetilde{\Omega}(\epsilon_2^3).$$

*Proof.* In this case, by Lemma 8 we have

$$\mathbb{E}[f(\mathbf{x}_{j+1}) - f(\mathbf{y}_j)] \leq -\frac{\epsilon_2^3}{24L_2^2}$$

For SCSG-Epoch [13], we have

$$0 \leq \mathbb{E}[\|\nabla f(\mathbf{y}_j)\|^2] \leq \frac{5L_1 b^{1/3}}{c' m_1^{1/3}} \mathbb{E}[f(\mathbf{x}_j) - F(\mathbf{y}_j)] + \frac{6G^2}{m_1}.$$

Hence,

$$\mathbb{E}[f(\mathbf{y}_j) - f(\mathbf{x}_j)] \leq \frac{6c'G}{5L_1 m_1^{2/3} b^{1/3}}$$

Thus,

$$\mathbb{E}[f(\mathbf{x}_{j+1}) - f(\mathbf{x}_j)] \leq -\frac{c^3 \epsilon_2^3}{12L_2^2} + \frac{6c'G}{5L_1 m_1^{2/3} b^{1/3}}$$

By setting $m_1 \geq (144GL_2^2 c'/(5c^3 L_1 b^{1/3} \epsilon_2^3))^{3/2}$, we have

$$\mathbb{E}[f(\mathbf{x}_{j+1}) - f(\mathbf{x}_j)] \leq -\frac{c^3 \epsilon_2^3}{24L_2^2} = -\widetilde{\Omega}(\epsilon_2^3)$$

$\square$

Combining Lemma 10 and Lemma 11 and following the analysis of Theorem 14 in [11], within $\widetilde{O}\left(\max(\frac{b^{1/3}}{\epsilon^{3/4}}, \frac{1}{\epsilon_2^3})\right)$ outer iterations, there exists at least one $\mathbf{y}_j$ such that $\mathbf{v}_j^\top H_{\mathcal{S}_2}(\mathbf{y}_j)\mathbf{v}_j \geq -\epsilon_2/2$ and $\|\nabla f(\mathbf{y}_j)\| \leq \epsilon_1$ with high probability. As a result, at such a $\mathbf{y}_j$ AdaNCD-SCSG terminates with a high probability as long as $|\mathcal{S}| \geq \widetilde{O}(1/\epsilon^2)$ for the stopping criterion to pass. Similar to the proof of Theorem 2, upon termination, we have $|\nabla f(\mathbf{y}_j)| \leq 2\epsilon_1$ and $\lambda_{\min}(\nabla^2 f(\mathbf{y}_j)) \geq -2\epsilon_2$, which completes the proof.