[Reviews · NeurIPS 2018]

Reviewer 1



The paper is correct and OK, but the idea is not that novel.

Reviewer 2



This paper proposed a series of adaptive negative curvature descent strategies to approximate the smallest eight-value of Hessian by the current gradient’s magnitude. Both deterministic and stochastic non-convex optimization have been considered. The algorithm terminate condition and time complexity were reported. The idea seems interesting and make sense for me. However, I am afraid the details of this work is not clearly stated. For example, in Algorithm 5, if the condition in step 5 is not satisfied, who will be return? The experimental part is also weak. I am afraid only reporting the oracle calls are not convincing. Other measures, average statistical results, or visual results can be considered to illustrate the superior of their algorithms. Furthermore, why the baseline method NCD are not reported in stochastic case in Figure 1? Finally, some presentation issues should also be fixed. For example, in line 9 of abstract, the state “we propose an adaptive NCD to allow for an adaptive error dependent on …” should be revised.

Reviewer 3



This paper investigates methods that use approximate negative curvature directions to accelerate optimization of non-quadratic convex functions. The negative curvature direction is the eigenvector corresponding to the smallest eigenvalue, which for saddle points and concave points are negative. A difficulty in such problems is that finding this eigenvector requires repeated hessian/vector products via power method or Lanczos method, which is much more expensive than simple gradient methods. Therefore, a key focus of this paper is to study the convergence behavior when only noisy curvatures can be computed, e.g. Lanczos or some other algorithm terminated early. The main novelty of the paper is that at each iteration, a noisy curvature is computed, and if the direction is "negative enough", that direction is chosen. If not, a standard gradient step is chosen. This allows for analysis of much noisier negative curvature steps. Major concerns: There should be some empirical comparisons against known methods (so far I only see comparisons against variants if their own proposed methods). To clarify, Lanczos methods ARE using repeated Hessian/vector products. The language in the main text seems to suggest the proposed method is doing something different, when in fact both this and Oja's method are doing basically the same thing. Overall I think this is a nice paper with nice contributions (although I am not an expert at the latest work in this area) but the writing could be tightened up a bit, to ensure nothing is misleading or unclear. Proofs: The argument for proposition 1 is entirely circular; the result is basically restated in line 1 of the proof. It's possible I'm missing something but for the proof of lemma 1: for clarification, add assumption that ||v||=1? also I think line after 61, the first equality should be inequality, and 8 should be 4. (minor) Minor suggestions: line 146: there should be a gradient sign line 154: we don't really need z. It never features.

Reviewer 4



This paper utilizes the Negative curvature descent (NCD) method to design adaptive algorithms for non-convex optimization aiming at finding second-order stationary points or local minima. The theoretical convergence rate of the adaptive NCD matches the state-of-the-art NCD-AG. Supporting experiment results using cubic regularization, regularized non-linear least-square, and one hidden-layer NN implies a significant supercedes the NCD-AG. In some sense, this work is a continuation of the work by Carmon et al (2016) in the deterministic setting. This work made a key contribution that at each update of AdaNCD\textsuperscript{det}, it chooses one of the GD (gradient-descent) step and NCD (Negative curvature descent / Lanczos) step, whichever decreases the function value more. The benefit of this method is that one only need to solve NCD with an accuracy of $O(\sqrt{\epsilon} \lor \|\nabla f(x)\|^{1/2})$ instead of $O(\sqrt{\epsilon})$. It is unsurprising to me (and also argued in the paper) that the convergence rate matches the state-of-the-art $O(\epsilon^{-1.75})$ gradient complexity when finding an $(\epsilon,\sqrt{\epsilon})$-approximate SOSP, which is the state-of-the-art local-minimizer finding algorithm [Carmon et al 2016, Agarwal et al 2017]. I tend to vote yes for this paper, not because of its technical proof method which has largely appeared in earlier literatures, but because the simple adaptive twist that makes it numerically work in practice. Most non-convex optimization theory works notoriously lack of such numerical support, however, making them numerically unattended. The work also dealt with the case when $\epsilon_2 = \epsilon^\alpha$ with $\alpha\in (0,1)$, and stochastic setting. Perhaps, in future work, the author also want to include the analysis of the finite-sum setting. Some minor comments: In Algorithm 5, the authors might want to explain why one can solve AdaNCG at accuracy $\epsilon^{3/4}$ and then switch gear to the AGD iteration. Should the accuracy be $\epsilon$ so they match up? The authors might want to carefully name the algorithms so as not to cause any confusion. For example, Algorithm 1, 3 and 5 have similar names, but the second (resp.~third) serves as a subroutine of the first (resp.~second). For the stochastic setting, the AdaNCD+SCSG result seems very similar to the so-called Neon+SCSG method (Xu et al 2017; Allen Zhu and Li, 2017). Does the complexity in Theorem 4 matches up with Neon2 in Allen-Zhu and Li (2017)?